# REPRESENTING SPEECH THROUGH AUTOREGRESSIVE PREDICTION OF COCHLEAR TOKENS

## ABSTRACT

We introduce a biologically-inspired model for encoding speech through an autoregressive prediction objective on input representations modeled after the human cochlea. Our modeling framework is inspired by the human auditory processing hierarchy. The first stage of our framework transforms the raw audio waveform into a time-frequency representation based on the human cochlea, with an intermediate stage that bottlenecks the audio into discrete units which we denote as *cochlear tokens*. The second stage of our framework learns a simple, yet powerful, autoregressive sequence model over the cochlear tokens. We demonstrate that our model learns meaningful representations of phonemes and word identities, and state-of-the-art representations of lexical semantics. In addition, our model shows competitive performance on a diverse set of downstream speech tasks from the SUPERB benchmark. Complementing our model's strong representational capabilities, we demonstrate its abilities to generate continuations of audio at various temporal scales, which can be visualized in a spectrogram space to provide insights into the model's predictions. Our model provides a novel framework for speech representation learning, aiming to advance the development of more human-like models that flexibly and efficiently handles a range of speech-based tasks.

## 1 INTRODUCTION

Humans possess a remarkable ability to perform a wide range of distinct tasks from speech–ranging from recognizing words in noisy environments and separating multiple speakers to identifying who is speaking and interpreting the emotional tone of their voice. These highly distinct processes are carried out by the human ear and networks of biological neurons. However, building artificial neural network models that replicate the human ability to flexibly and efficiently understand and interact with the world through speech in highly diverse ways remains a significant challenge (1; 2; 3). **To bridge this gap, we introduce CochStream, a biologically-inspired model that learns versatile speech representations through a simple and scalable autoregressive prediction objective on a time-frequency representation inspired by the human cochlea (4; 5; 6; 7).**

### 1.1 SPEECH REPRESENTATION LEARNING

Speech representation models, also known as audio encoders, broadly take an audio signal as input and embed it in a series of discrete tokens or continuous embeddings for use in a variety of downstream audio tasks (3). Several approaches exist for speech representation learning. One popular approach uses neural audio codecs, which learn compressed audio representations by preserving a minimal amount of information needed to reconstruct an audio signal. This compression allows codec models to naturally recover the original signal from the learned codes (8; 9; 10; 11; 12; 13; 14; 15; 16). Many audio codec models are specific to speech (14; 17; 13; 15) while others are more general and include other audio categories such as music, environmental sounds, etc. 10; 8. Models such as SoundStream (8) achieve impressive bit rates (as low as 3kbps) while maintaining high reconstruction quality. These audio codes can then be used as representation for downstream audio tasks (18; 19; 2; 15). However, although these models retain high-fidelity information about the acoustic details due to the reconstruction objective, learning the appropriate acoustic invariances continues to be a challenge (2). Further, high-fidelity signal reconstruction is not a biologically plausible objective – humans

show robust invariance to many low-level features which for instance allows us to understand diverse speakers (20; 21).

A second popular approach for speech representation learning is prediction-based modeling. These models predict features derived either from the raw waveform (22; 23; 24) or from a time-frequency representation of the audio (25; 26; 27; 28). Broadly, these prediction-based speech models fall into two categories: those that predict future frames with an autoregressive objective (25; 26; 27; 28) or those that predict masked frames from surrounding frames (24; 29; 22; 30) (analogous to the causal and bi-directional prediction approaches in language modeling). The resulting representations are then utilized in various downstream audio tasks, for instance language modeling (30; 22; 31; 32; 33; 34). One of the most widely used predictive models is HuBERT (22) which adapts the bi-directional BERT (35) objective for speech representation learning via waveform features.

A third common approach is contrastive learning, where frames from different audio samples are pushed together or pulled apart in the embedding space based on a specified objective. One popular model within contrastive models is wav2vec2 (36) which contrasts masked-out audio segments from distractors in combination with an auxiliary objective. Other models that utilize contrastive objectives include CPC (37), SCPC (38), and COLA (39). Broadly, this approach can yield powerful representations but requires a heuristic to determine positive and negative samples, implicitly enforcing which aspects of the audio signal are retained. Moreover, contrastive objectives often rely on directly contrasting embeddings across hundreds or thousands of diverse samples simultaneously, which, arguably, is not a biologically plausible operation.

Although these three speech representation learning strategies are distinct, their objectives can be combined and augmented with additional heuristics. For instance, a state-of-the-art model, WavLM (23), combines the HuBERT bi-directional prediction objective (22) with a noisy input transformation to obtain strong numbers on the speech representation SUPERB benchmark (1). However, as with most ensemble models, these performance gains come at the cost of additional hand-crafted complexity.

### 1.2 Our Approach: A Two-Stage Framework for Autoregressive Prediction on Biologically-Inspired Input Representations

In contrast to these past approaches, our proposed framework does not rely on signal-reconstruction objectives (like neural codec models), non-causal prediction objectives (like bi-directional prediction models), or intra-batch contrasting of samples (like many contrastive models). Instead, our framework takes inspiration from the human auditory processing stream and operates in two stages:

The first stage of our framework transforms the raw audio waveform into a time-frequency representation based on the human cochlea (**WavCoch**; Figure 1A-C). Note that this approach in some way resembles neural audio codecs, but instead of reconstructing the *same* signal, we predict another audio representation–one known to be computed within the human auditory processing hierarchy—the time-frequency cochleagram (4; 5; 6; 7). We probe the representations in an intermediate bottleneck stage of WavCoch which effectively discretizes the audio representations (Section 2.1.1). We refer to these discrete, intermediary representations as **cochlear tokens** (Figure 1B), and broadly term this strategy of training a computational model to transition between representational states while extracting intermediary representations as *"Transformation Imitation"* (see Section 4). We hypothesize that the cochlear tokens learned through this transformation will develop inductive biases similar to those of the human cochlea, benefiting the model's ability to process diverse speech patterns efficiently (e.g., 40).

The cochlear tokens serve as input to the second stage of our framework, **CochStream**, which is a simple autoregressive sequence model, trained to predict the upcoming cochlear token (Figure 1D) (Section 2.1.2). Since the cochlear tokens were derived from a waveform-to-cochleagram transformation, these predicted cochlear tokens can naturally be decoded into the cochleagram representation for inspection and interpretability (Figure 1E). Compared to existing autoregressive prediction models, we do not jointly learn to embed and predict future frames, but rather utilize our biologically-inspired WavCoch quantization model to identify tokens, and subsequently learn to model a distribution over this fixed token vocabulary via autoregressive prediction.

Hence, we formulate speech representation learning through a simple, yet powerful, autoregressive prediction objective on biologically-realistic input representations, cochlear tokens. We demonstrate

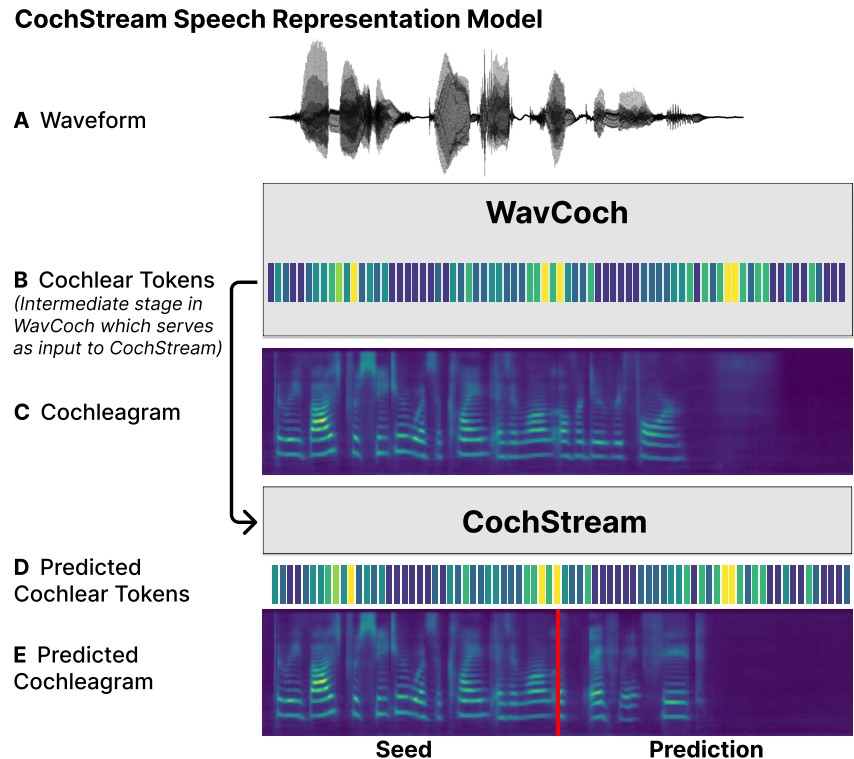

Figure 1: **Schematic of CochStream speech representation model**. Description of the steps associated with the model can be found in the Introduction (1.2) and Methods (2.1).
.

that our framework leads to the emergence of representations from which phonemes, word forms, and word meanings (lexical semantics) can be decoded at competitive levels, when compared to several comparison models (Section 3.1 and 3.2), with our approach obtaining state-of-the-art performance on lexical semantics (Section 3.1). Further, we show that our learned representations serve as a a powerful backbone for various downstream speech tasks, as evaluated on the SUPERB benchmark (1) (Section 3.3). Finally, unlike the comparison models, we demonstrate CochStream's capability to generate continuations of audio when prompted, which can be visualized in a cochleagram time-frequency space to provide insights into the model's predictions (Section 3).

**So, why should we care about our novel framework? From a machine learning perspective**, we introduce cochlear tokens which fit within the context window of a standard Transformer, effectively leveraging the power of autoregressive modeling (41). By casting speech representation learning as a generic sequence prediction task, our model matches the performance of comparably sized models. Yet, with the scalable nature of autoregressive models (demonstrated in Section 3.1 and 3.3) and availability of large amounts of unlabeled speech data, our approach is well-positioned to outperform existing methods at larger scales. **Within cognitive science and neuroscience**, a growing body of research uses artificial models to investigate human behavior and neural processes, under the premise that models operating on realistic input signals and/or solving biologically relevant tasks may develop representations or computations resembling those of biological systems (42; 43). Recent work has investigated whether these models mimic human behavior (44; 40; 45; 46; 47) and neural activity (48; 49; 50; 51; 52). However, current artificial models fall short in matching key aspects of biological systems (53; 54; 7). Hence, developing new classes of models that incorporate biological heuristics, such as a cochlear-inspired input representations, is crucial for advancing accurate computational accounts of human cognition and brain function.

## 2 METHODS

### 2.1 COCHSTREAM MODEL

The CochStream model (Section 2.1.2) is trained to autoregressively predict a sequence of cochlear tokens produced by the WavCoch encoder (Section 2.1.1).

### 2.1.1 INPUT REPRESENTATIONS: WAVCOCH

We propose a method for efficiently tokenizing audio through a waveform-to-cochleagram transformation model. This model, **WavCoch**, loosely mimics the function of the human cochlea, transforming auditory inputs into a time-frequency representation (4; 5; 6; 7).

WavCoch is a vector-quantized encoder which takes as input 5s clips of mono audio waveforms sampled at 16 kHz and is trained to predict a time-frequency representation of the corresponding audio clip. The target time-frequency representation is a human-inspired cochleagram representation (55; 56; 7), consisting of 221 frequency bins and 988 temporal steps (7). The purpose of the WavCoch model is to extract discrete tokens from continuous audio signal to serve as the input to CochStream. The full model diagram is illustrated in Appendix Figure 1 and the steps are described below.

First, the raw waveform (shape: [1,80000] for 5s of mono audio sampled at 16kHz) undergoes the Fourier Transform by computing Twiddle Factors (57). These factors represent complex sinusoidal components that decompose the signal into its frequency spectrum. The Twiddle Factors are applied to the audio signal through a 1D convolution (window size 1,001 and hop length 80 samples) which transforms the signal into the time-frequency domain. Second, each 5 ms temporal step of this time-frequency representation is fed into two fully-connected (FC) layers with ReLU nonlinearities (with 512 hidden units each). Third, these embeddings are then passed through a 13-dimensional LFQ bottleneck (58), which effectively binarizes the representation. We read out the activations of this bottleneck as a 13-bit binary code which can be interpreted as one of $2^13 = 8,192$ discrete tokens (we note that all models will be updated to use 13-bit codes in the final paper version. The current CochStream versions besides the LibriSpeech version use 14-bit codes). Fourth, the output of the LFQ bottleneck is then projected to a 211 dimensional output, through two 1-dimensional convolutional layers (kernel size 10 and stride 1), separated by ReLU nonlinearities. This output corresponds to the frequencies in the cochleagram representation (7) which it is supervised to match via L2 error. Thus for every 5 seconds of audio, WavCoch extracts a sequence of 988 integers in the range [0, 8192) through the LFQ bottleneck, denoted as **cochlear tokens**, to feed into CochStream.

### 2.1.2 SEQUENCE MODELING: COCHSTREAM

**CochStream** is a GPT-style autoregressive Transformer (59). We train two versions: CochStream-base (97M parameters), with 12 layers, 12 attention heads and an embedding size of 784 and CochStream-large (1.3B parameters) with 24 layers, 16 attention heads, and an embedding size of 2,048. Both models have a vocabulary size of 16,384. The CochStream model takes as input the cochlear token sequence produced by WavCoch (Section 2.1.1) and predicts the next token in the sequence. The context length is approximately 20s (4,096 tokens). We utilize a learned positional embedding and compute the cross-entropy loss between the predicted logits and the true next token in the sequence.

Additionally we train an ablation model called CochStream-large-ll which is trained on the Libri-Light (60) dataset of 60k hours of publicly available speech recordings. This model has 1B parameters with an embedding size of 1024, 12 heads and 48 layers to match baseline models (22), (23). This model utilizes a WavCoch quantizer trained on the librispeech960 (61) dataset of 960 hours of publicly available speech, and utilizes a vocabulary size of 8,192, (following our ablations in Appendix Section 1.5).

### 2.1.3 TRAINING

To train WavCoch, we use the AdamW optimizer (62) with a peak learning rate of 1e-4 and a 200k step cosine-decay schedule. We use a batch size of 512, and a weight decay of 0.1. The training data consisted of internet audio clips containing naturalistic speech, largely podcasts and lecture recordings sampled at 16 kHz. We train the model on 500 hours of such data.

To train CochStream, we also use the AdamW optimizer with a peak learning rate of 3e-4 and a 200k step cosine-decay schedule with a 2k step warmup. We use weight decay of 0.1 and norm 1.0 gradient clipping. We train the model with a batch size of 256. CochStream was trained on 50,000 hours of such data (no associated transcriptions).

### 2.1.4 OBTAINING COCHSTREAM EMBEDDINGS

We obtain CochStream embeddings by pooling the embeddings of all the tokens associated with the corresponding temporal section of the cochleagram via ground-truth phoneme or word boundaries (Section 3.1). For the pooling operation, we tested mean/max/min pooling for the linear probing experiments and lexical similarity for CochStream and the comparison models.

### 2.2 COMPARISON MODELS

CochStream is compared to three state-of-the-art speech representation models using the HuggingFace Transformers package (63): HuBERT-xl (identifier: *facebook/hubert-xlarge-ll60k*) (22), wav2vec2-large (identifier: *facebook/wav2vec2-large*) (64), wavLM (identifier: *microsoft/wavlm-base*, and *microsoft/wavlm-large*) (23). For the SUPERB benchmark, we additionally compare against two smaller models which share some similarity to our method, specifically, APC (26) and vq-wav2vec (65).

### 2.3 EVALUATION METRICS

To investigate the representational power of the CochStream embeddings, we linearly probe CochStream for phoneme identity, word identity, and lexical semantics. Additionally, to validate the use of CochStream as a powerful generic speech representation backbone, we evaluate it on the SUPERB benchmark (1), which fits specialized downstream task models on top of frozen backbone representations.

### 2.3.1 PHONEME/WORD LINEAR PROBING

To probe for phoneme and word identity representation, we use the TIMIT dataset (66) consisting of approximately five hours of audio recordings with ground-truth phoneme- and word-boundaries. We use the train and complete test sets with exclusion of the "SA" sentences to allow for train and test sets that are non-overlapping in sentences as well as speakers. For phoneme classification, we followed the standard protocol of collapsing the TIMIT phoneme labels from 60 to 39 classes (67). The number of words in the TIMIT train set is 30,132 and 8,128 for the test set. The number of phonemes in the TIMIT train set is 140,225 and 50,754 in the test set. For linear probing, we use the scikit-learn LogisticRegression multiclass classifier (max_iter = 10000, solver = lbfgs, penalty = l2) (68). For each model, we identify the best-performing layer via mean pooling of the embeddings associated with each phoneme/word. The reported values are weighted accuracy scores (as the classes are imbalanced). To determine chance performance, we compute the probability of the most likely class label.

### 2.3.2 LEXICAL SEMANTIC SIMILARITY (SSIMI)

We use the "sSIMI" lexical semantics benchmark developed for the ZeroSpeech 2021 challenge (69). The benchmark was constructed to probe whether audio-based models learn lexical semantics. The benchmark consists of pairs of words with ground-truth human similarity judgments (on a 0 and 10 scale) collected from behavioral experiments. For instance, a pair of words such as "water" and "river" have a human similarity score of 9.8, while a pair like "festival" and "whiskers" have a score of 0.2. Two audio subsets exist for these pairs of words: i) a natural dataset, consisting of the pairs of words present in LibriSpeech (70), and ii) a synthetic datasset, consisting of all pairs. The two datasets are each separated into a dev and test section (the LibriSpeech dataset is contains 309 and 3,753 word pairs for dev and test, and 705 and 9,744 for the synthesized subset). Each audio clip contains just one word, from which we extract embeddings. Following the procedure in (69), for each model, we identify the optimal embedding pooling strategy (mean/max/min) as well as layer based on the dev set, and obtain the final scores on the test set. The final sSIMI score is computed as

| Dataset | Model Size | Dataset Size | Phoneme Decoding | | Word Decoding | |
|---|---|---|---|---|---|---|
| | Params | Hours | Accuracy ↑ | Random ↑ | Accuracy ↑ | Random ↑ |
| HuBERT-xl | 1,000M | 60K | **0.93** | 0.20 | **0.88** | 0.07 |
| HuBERT-base | 97M | 1K | 0.85 | 0.20 | 0.77 | 0.07 |
| wav2vec2-large | 317M | 60K | 0.79 | 0.20 | 0.43 | 0.07 |
| WavLM-large | 317M | 94K | 0.91 | 0.20 | 0.85 | 0.07 |
| CochStream-base | 97M | 0.5K | 0.82 | 0.20 | 0.48 | 0.07 |
| CochStream-large | 1,300M | 50K | 0.92 | 0.20 | 0.67 | 0.07 |
| CochStream-large-ll | 1,000M | 60K | 0.92 | 0.20 | 0.69 | 0.07 |

Table 1: **Linear probing performance for phonemes or words on the TIMIT dataset**. Reported values are weighted accuracy scores on the TIMIT test set, consisting of non-overlapping sentences uttered by non-overlapping speakers from the train set.

the Spearman correlation between the cosine distance of embeddings for pairs of words and the true human similarity scores.

### 2.3.3 SPEECH PROCESSING UNIVERSAL PERFORMANCE BENCHMARK (SUPERB)

To comprehensively evaluate the strength of the representations produced by our model, we evaluate it on the SUPERB benchmark which contains 15 tasks, categorized into five aspects of speech: content, speaker, semantics, paralinguistics, and generation. We report values on a subset of 7 tasks spanning all 5 categories. We refer to the original paper for additional details on the benchmark (1).

## 3 RESULTS

### 3.1 COCHSTREAM EMBEDDINGS CONTAIN INFORMATION ABOUT PHONEME IDENTITY, WORD IDENTITY, AND LEXICAL SEMANTICS

First, we investigate whether the representations from CochStream contain information about phoneme and word identity. The core premise is that if we can linearly decode these properties from the audio representations they will serve as useful representations for downstream tasks that require this information. We fitted linear classifiers on the phonemes/words in the train set of TIMIT (66) and tested the classifiers on the test set consisting of non-overlapping sentences and speakers (Section 2.3.1). We compared CochStream to three state-of-the-art models, HuBERT-xl (22), wav2vec2-large (64), and WavLM-large (23). As evidenced in Table 1, CochStream showed competitive performance compared to the comparison models. For phoneme decoding, CochStream surpassed wav2vec-large by a large margin and performed on par with HuBERT-xl and WavLM-large. The error patterns of CochStream were sensible. For instance, for phoneme probing, "er" was often confused with "r', or "ah" with "ih" (see Appendix I). For word decoding, CochStream once again surpassed wav2vec-large, however, CochStream fell short of HuBERT-xl and WavLM-large. We hypothesize that the subpar word decoding performance of CochStream relative to HuBERT and WavLM may be attributed to the the fact that HuBERT and its derivative models (WavLM) were exposed to global clustering operations aimed at discovering word-like units. In contrast, CochStream did not undergo any such global operations. Lastly, we emphasize the decoding performance for both phonemes and words scales well with CochStream size.

Second, we turn to a benchmark that goes beyond identity of phonemes/words, but instead asks whether the speech models learn representations of word meanings (lexical semantics). This benchmark (sSIMI; Section 2.3.2) evaluates the correlation between embeddings derived from audio associated with pairs of words (e.g., "water" and "river") to that of ground-truth human similarity judgments (Section 2.3.2) (69). The performance of speech models on this benchmark have been described as "modest" (69; 71), suggesting that this class of models struggle in learning semantics independently from the acoustic properties of words. Table 2 presents the performance of CochStream alongside the baseline models. The embeddings of CochStream showed superior performance on lexical semantics on both the natural and synthetic data subsets compared to the other models. Further,

| Dataset | Model Size | Dataset Size | LibriSpeech Audio | Synthetic Audio |
|---|---|---|---|---|
| | Params | Hours | Accuracy ↑ | Accuracy ↑ |
| HuBERT-xl | 1,000M | 60K | 7.81 | 10.37 |
| HuBERT-base | 97M | 1K | 6.10 | 7.48 |
| wav2vec2-large | 317M | 60K | 6.41 | 7.19 |
| WavLM-large | 317M | 60K | 10.50 | 10.41 |
| CochStream-base | 97M | 0.5K | 10.63 | 10.12 |
| CochStream-large | 1,300M | 50K | **12.52** | **10.64** |
| CochStream-large-ll | 1,000M | 60K | 10.99 | 10.52 |

Table 2: **Semantic similarity scores on the ZeroSpeech 2021 Lexical Semantic Benchmark**. Reported values are Spearman correlations (multiplied by 100 per (69)) between the embeddings for pairs of words versus human similarity judgments on the same pairs of words. The scores were obtained on the test sets of two dataset subsets (LibriSpeech and a synthetic set, see Section 2.3.2).

| Setting | Model | Dataset | PR | ASR | IC | KS | SID | ER | SS |
|---|---|---|---|---|---|---|---|---|---|
| | Params | Hours | PER ↓ | WER ↓ | Acc↑ | Acc↑ | Acc↑ | Acc↑ | SI-SDRi ↑ |
| HuBERT-large | 1,000M | 60K | 3.53 | 3.62 | 98.76 | 95.29 | 90.33 | 67.62 | 10.45 |
| wav2vec2-large | 317M | 60K | 4.75 | 3.75 | 95.28 | 96.66 | 86.14 | 65.64 | 10.02 |
| vq-wav2vec | 32M | 1K | 33.48 | 17.71 | 85.68 | 93.38 | 38.80 | 58.24 | 8.16 |
| WavLM-base | 97M | 1K | 4.84 | 6.21 | 98.42 | 96.79 | 84.51 | 65.94 | 10.37 |
| WavLM-large | 317M | 94K | 3.06 | 3.44 | 99.31 | 97.86 | 95.49 | 70.62 | 11.19 |
| APC | 4M | 0.36K | 41.98 | 21.28 | 74.69 | 91.01 | 60.42 | 59.33 | 8.92 |
| CochStream-base | 97M | 0.5K | 5.62 | 6.04 | 98.10 | 95.29 | 78.30 | 64.42 | 10.00 |
| CochStream-large | 1,300M | 50K | 4.20 | 3.87 | 98.10 | 96.12 | 82.14 | 68.37 | 10.67 |
| CochStream-large-ll | 1,000M | 60K | 4.16 | 3.78 | 97.85 | 95.72 | 80.02 | 68.98 | 10.66 |

Table 3: **Model embedding performance on various downstream tasks (SUPERB)**. Reported values are obtained by training a downstream task decoder on top of a frozen model backbone (1).

the CochStream model scores increased with model size. In sum, we demonstrate that CochStream embeddings learn state-of-the-art representations for lexical semantics.

## 3.2 PHONEME, WORD, AND LEXICAL SEMANTIC REPRESENTATIONS ACROSS TIME AND LAYERS

We investigate how information about phonemes, words, and lexical semantics (Tables 1 and 2) is distributed across multiple layers in CochStream. In addition, we investigate the extent to which phoneme/word information is distributed across time. To do so, we run probing experiments on CochStream embeddings that are shifted in time (i.e., "intentionally" mismatched; Figure 2A bottom panel). Figure 2A shows that the highest accuracy on phoneme/word decoding was obtained at the true temporal window, with decreasing performance when the windows are shifted either left or right. Layer 1 had the lowest performance, with highest performance from middle layers. Figure 2B shows the lexical semantic benchmark, and in contrast to Figure 2A, the last layers of the model tend to exhibit the highest correlation with human similarity judgments.

## 3.3 COCHSTREAM SERVES AS A STRONG FROZEN BACKBONE FOR DOWNSTREAM AUDIO TASKS

After having established that the CochStream representations themselves contain meaningful phoneme, word, and lexical semantics information (Section 3.1 and 3.2), we investigated whether the frozen representations of CochStream would serve as powerful features for training decoders across a range of various audio tasks. To do so, we leveraged 7 different SUPERB tasks, spanning the 5 broad categories present in this benchmark (1). As illustrated in 3 CochStream-large outperformed two of the most similar models evaluated on the benchmark – APC and vq-wav2vec, while performing competitively against state-of-the-art models on most other tasks. In particular, CochStream showed very strong performance on automatic speech recognition (SS), intent classification (IC), and speech

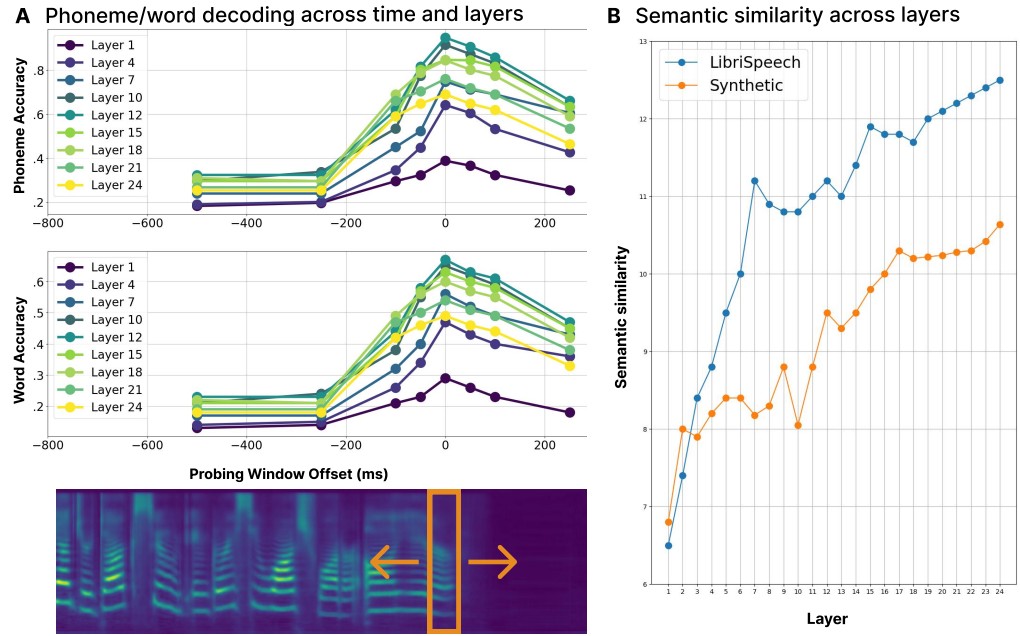

Figure 2: **A. Linear probing accuracy on TIMIT phonemes/words across time and layers.** The scores were obtained across model layers, with various shifts in temporal windows relative to the true phoneme/word from which the embeddings are extracted (x-axis) as demonstrated in the bottom panel. **B. Semantic similarity scores across layers on the ZeroSpeech 2021 benchmark.** Lines denote performance on the test set of the two data subsets.

separation (SS). In contrast, CochStream-large had subpar performance on speaker identification (SID) compared to the other models in the same parameter class. These findings illustrate that the features learned by CochStream serve as versatile representations for diverse downstream audio tasks, and importantly, demonstrate scalability (compare CochStream-base with CochStream-large).

### 3.4 COCHSTREAM LEARNS SHORT- AND LONG-RANGE SPEECH STATISTICS

Having established that CochStream representations contain decodable information of phonemes, words, and lexical semantics (Section 3.1), we ask: In the absence of ground-truth phoneme/word boundaries, does CochStream learn the statistics of speech? To answer this question, we leverage the fact that CochStream was trained to perform predictions in a space that can be visualized and interpreted, that is, the time-frequency cochleagram image (we note that models like HuBERT and wav2vec lack this capability). We hypothesize that if CochStream has learned the statistics of speech, it should manifest as two distinct modes: On short timescales, when provided enough conditioning (such as the first part of a common word), the model should be able to complete the cochleagram in a manner that is consistent with the remainder of the word. Conversely, at longer timescales, the model should diverge in the generated outputs, as many plausible words can follow any given phoneme or word. We provide CochStream with the beginnings of audio clips from the TIMIT test set (out-of-distribution for CochStream) and qualitatively analyze the resulting predictions (Figure 3).

To first test whether our model learns speech structure on short timescales, we provide CochStream with information about the first part of a common word (e.g., "she", consisting of the phoneme "sh"), and evaluate its ability to predict the continuations from this first phoneme across different speakers in the TIMIT test set. As evidenced in Figure 3A, the model learns to consistently complete the phoneme with an "iy" phoneme, resulting in the word "she". Conversely, if a phoneme has several likely continuations, the model learns to complete the phoneme with different words. As an example, we seed CochStream with the first phoneme ("wa") of the word "water" and "wash" from two different speakers (Figure 3B). CochStream sometimes predicts the remainder of the true

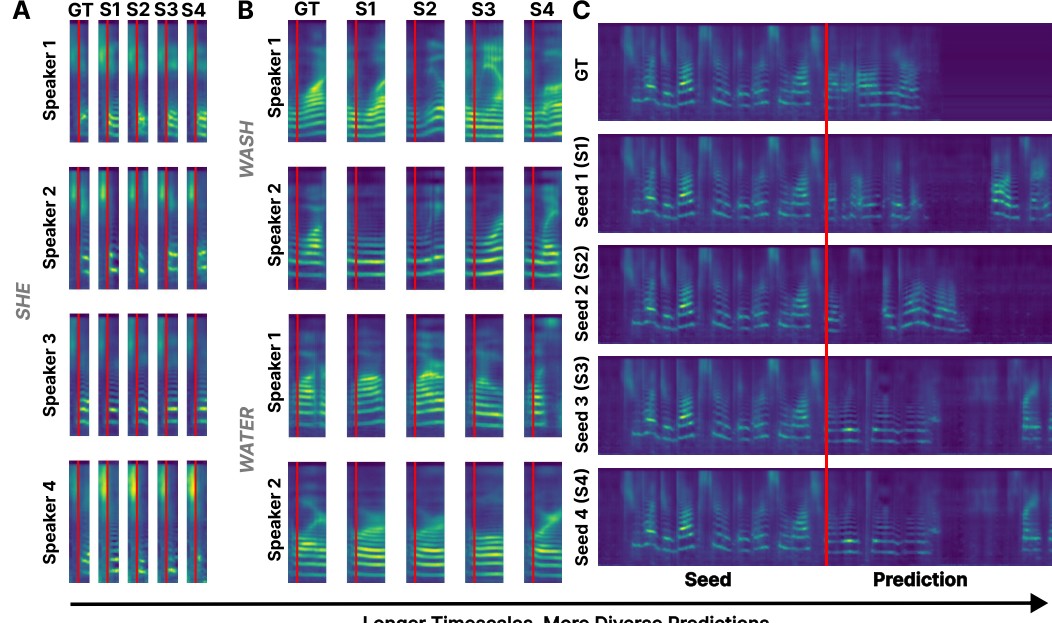

Figure 3: **Prediction of cochleagrams produced by the CochStream-base model: A.** CochStream is seeded with the first phoneme of the word "she" (left of red vertical line) and predicts the word completion (right of red line). The ground-truth (GT) cochleagram is shown in the first column. **B.** CochStream is seeded with the first phoneme of the words "wash" and "water". **C.** CochStream is seeded with the first 2.5 seconds of an audio clip (red vertical line) from the TIMIT test set and predicts the remaining part of the clip across four different seeds.

word, while sometimes it predicts a counterfactual word completion, which is still consistent with the first phoneme. For example, we observe that the prediction under Seed 3 of Speaker 1's utterance "wash" looks more similar to Speaker 2's ground-truth utterance of the word "water" than its own ground-truth word ("wash"). Hence, these visualizations suggest that CochStream learns the statistics of how phonemes compose to words.

Second, to test for diversity of longer-range predicted output, instead of simply seeding the model with a phoneme, we seed the model with the first 2.5 seconds of a TIMIT audio clip (Figure 3C). We observe that the model is capable of predicting several seconds of plausible completions. Furthermore, we highlight that various non-cherry-picked random seeds yield highly different predictions without collapsing. See additional examples here: https://anonymous.4open.science/w/cochstream-project-page-0546/.

## 4 DISCUSSION AND LIMITATIONS

In this paper, we proposed a self-supervised speech representation model, CochStream, for encoding speech without the need for any ground-truth text annotations.

We demonstrated that CochStream learns representations from which phonemes, word forms, and word meanings (lexical semantics) can be decoded at competitive levels, with state-of-the-art performance on lexical semantics (Section 3.1). Further, we demonstrated that CochStream serves as a strong representational backbone for various audio tasks, such as automatic speech recognition or speech separation (Section 3.3). For both types of tasks, we observed predictable increases in task performance as we scale the model. Finally, we demonstrate that our modeling framework is robust to the training dataset used (Section 3.3), emphasizing that the core performance contribution of our work comes from our novel two-stage modeling framework (WavCoch tokenization followed by auto-regressive CochStream prediction).

One strength of our framework is the use of cochlear tokens derived from a human-inspired waveform-to-cochleagram transformation (WavCoch), which serve as the input representations to a sequence model (CochStream). Taking a step back: biological systems have hierarchical stages that transform one known representation into another (known) representation. Here, we leverage this knowledge of representational states to train WavCoch and explicitly *probe this transformation* to extract cochlear tokens for downstream use. This approach–which we denote as *"Transformation Imitation"*–holds great potential for extension to other perceptual domains. Another strength of cochlear tokens is their efficiency: each second of audio is represented with just 197 tokens. This downsampling makes the input inherently efficient, enabling real-time predictions. Importantly, our Transformation Imitation framework is adaptable to different representations; for example, the cochleagram could easily be replaced with a standard mel-spectrogram. While we do not make explicit claims about the superiority of the cochleagram representation over the mel-spectrogram, our exploratory findings indicate that cochleagrams perform at least as well as spectrograms (as estimated through codebook usage and phoneme purity, see Appendix Section 1.5). Finally, we acknowledge that previous work has leveraged cochleagrams in speech representation learning (27; 72; 7) and various neural codec approaches are similar in spirit to WavCoch in the fact that they extract intermediate bottleneck representations for downstream tasks (8; 11; 12; 13; 14; 15; 16). However, we still believe that WavCoch is novel in its way of learning to transform one representation into *another* representation through a discrete quantization bottleneck (instead of auto-encoding, as done in related approaches).

A final advantage to emphasize is that CochStream allows to visualize and interpret resulting audio predictions directly (for example, Figure 3), a capability that many audio-based models do not have, making CochStream less of a "black box". Although the predicted outputs of CochStream cannot directly be transformed back into audio, we have performed initial analyses to generate audible responses see Appendix Section 1.6).

A couple limitations of our work exist. One limitation is that our model is trained on English speech, constraining the analyses we perform to tasks and materials in English (73; 74). We do emphasize that our model, as well as textless natural language processing more broadly, can serve as particularly useful for applications within low-resource languages. Another limitation is that due compute and time constraints, our models could not be scaled to their full potential.

In addition to CochStream's advantages for machine learning, our hope is that it will serve as a valuable model for the emerging field of "NeuroAI" (75). This field leverages artificial neural networks to better understand human behavior and brain function, testing whether artificial models trained on naturalistic data and/or biologically relevant tasks develop human-like solutions. Recent work has explored whether these models mimic human behavioral characteristics (44; 40; 45; 46; 47), as well as whether their internal representations align with neural activity in the human brain (48; 49; 50; 51; 52; 76).

We by no means claim that CochStream is a perfect biologically-inspired model, but it is a critical step in the right direction. CochStream uses a Transformer architecture—a choice driven by its proven efficacy for sequence-based tasks—which might have properties that make it incompatible with biological neurons (however, see 77; 78; 79 for how Transformer mechanisms could be implemented in biological hardware). Nevertheless, CochStream marks a significant step forward: it leverages biologically-realistic input representations and employs a simple, causal prediction mechanism without relying on additional hand-crafted heuristics. This approach contrasts with many speech representation models that depend on high-fidelity signal reconstruction objectives, bidirectional non-causal prediction, and the contrasting of hundreds of samples within a training batch (see Introduction 1).

In conclusion, we present a novel framework for speech representation learning, aiming to advance the development of more human-like, neurally plausible models that capture the intricacies of human speech processing.

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
