# APPENDIX: REPRESENTING SPEECH THROUGH AUTOREGRESSIVE PREDICTION OF COCHLEAR TOKENS

# I APPENDIX

## I.1 MODEL ARCHITECTURE

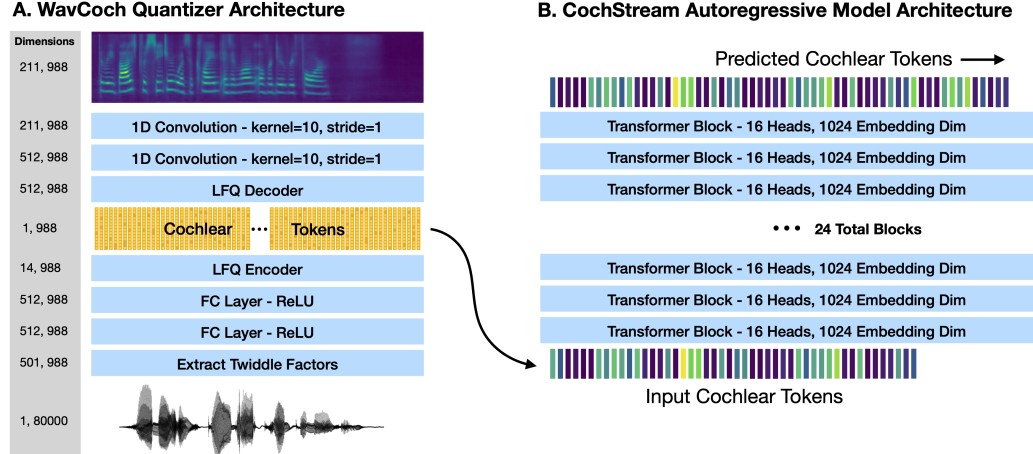

Figure 1: **A. WavCoch Quantizer Architecture:** First, the raw waveform (shape: [1,80000] for 5s of mono audio sampled at 16kHz) undergoes the Fourier Transform by computing Twiddle Factors [1]. These factors represent complex sinusoidal components that decompose the signal into its frequency spectrum. The Twiddle Factors are applied to the audio signal through a 1D convolution (window size 1,001 and hop length 80 samples) which transforms the signal into the time-frequency domain. Second, each 5 ms temporal step of this time-frequency representation is fed into two fully-connected (FC) layers with ReLU nonlinearities (with 512 hidden units each). Third, these embeddings are then passed through a 14-dimensional LFQ bottleneck [3], which effectively binarizes the representation. We read out the activations of this bottleneck as a 14-bit binary code which can be interpreted as one of $2^{14} = 16,384$ discrete tokens. Fourth, the output of the LFQ bottleneck is then projected to a 211 dimensional output, through two 1-dimensional convolutional layers (kernel size 10 and stride 1), separated by ReLU nonlinearities. This output corresponds to the frequencies in the cochleagram representation [2] which it is supervised to match via L2 error. Thus for every 5 seconds of audio, WavCoch extracts a sequence of 988 integers in the range [0, 16384) through the LFQ bottleneck, denoted as **cochlear tokens**, to feed into CochStream. **B. CochStream Autoregressive Model Architecture:** The cochlear tokens obtained in WavCoch are passed to a GPT-style autoregressive Transformer [4], denoted as CochStream. We train two versions: CochStream-base (97M parameters), with 12 layers, 12 attention heads and an embedding size of 784 and CochStream-large (1.3B parameters) with 24 layers, 16 attention heads, and an embedding size of 2,048. Both models have a vocabulary size of 16,384. The CochStream model takes as input the cochlear token sequence produced by WavCoch and predicts the next token in the sequence. The context length is approximately 20s (4,096 tokens). We utilize a learned positional embedding and compute the cross-entropy loss between the predicted logits and the true next token in the sequence.

## I.2 CONFUSION MATRIX FOR PHONEME DECODING

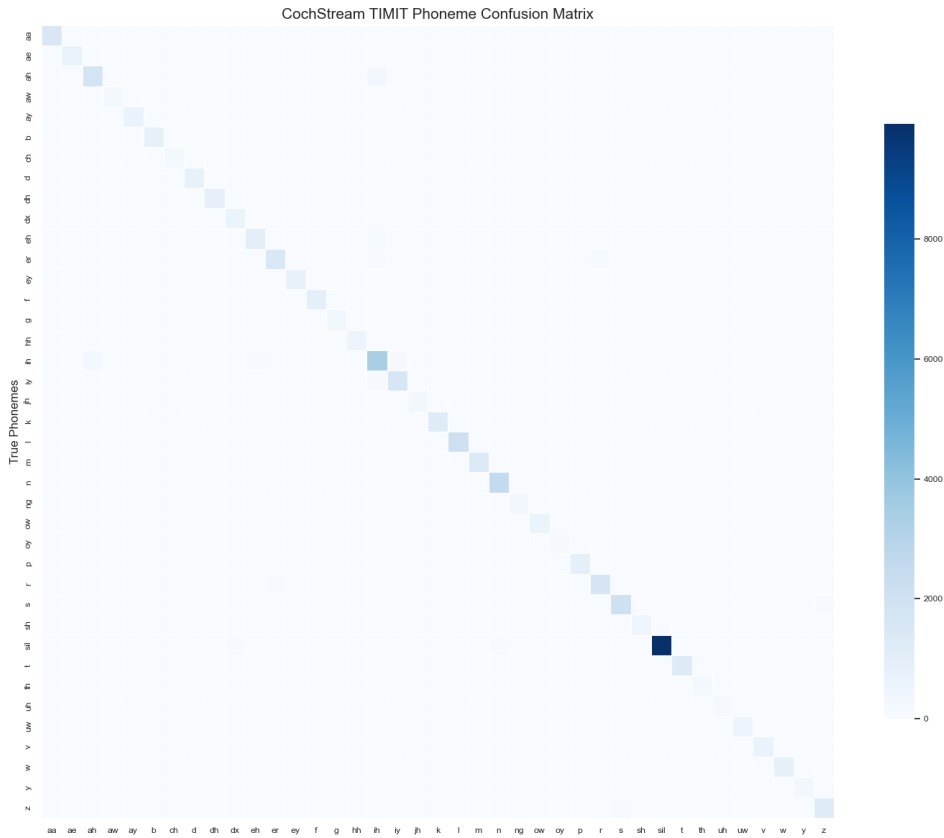

Figure 2: **Confusion matrix for phoneme decoding.** The plot shows performance of CochStream-large on the TIMIT test set.

## I.3 WAVCOCH VOCABULARY SIZE ABLATIONS

We perform ablations on the vocabulary size of the WavCoch model using the librispeech960 dataset. We train variants of WavCoch using a vocabulary size of 16,384, 8,192 and 4,096 (14, 13 and 12 bit codes respectively). For each of these models we evaluate reconstruction L2 error and phoneme cluster purity on an out of distribution test set (TIMIT test set). Phoneme cluster purity is defined as purity = (Count of most associated phoneme for token i) / (total counts for token i) providing an intuitive metric for how consistently a given token aligns with a specific phoneme. We report these result in Appendix Figure 3.

The plot shows that a vocabulary size of 8,192 presents both the highest cluster purity and lowest reconstruction error. Furthermore, this vocabulary size presents a local minima in the MSE space, and a maxima in the cluster purity space indicating it is the optimal size for generalization in this domain. We are very grateful to the reviewers' suggestion as it revealed an improvement in our model design. We will use a vocabulary of 8,192 for all future models.

## I.4 WAVCOCH TOKEN CLUSTERING

We analyze the distribution of tokens associated with specific phonemes in the TIMIT test set. We visualized the distribution in Appendix Figure 4 and found that multiple tokens are typically associated with each phoneme. Notably, a large number of tokens are linked to the "sil" (silence) label, but we would like to clarify that this label includes any non-speech sound, not true silence.

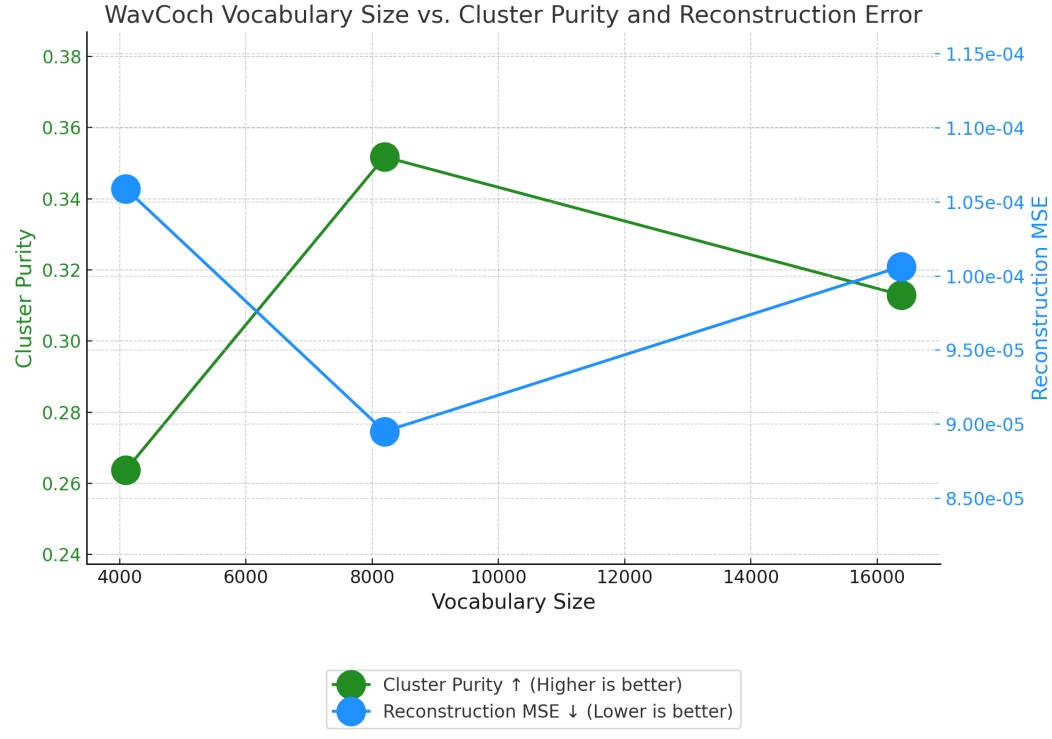

Figure 3: **Out of distribution generalization of WavCoch models with various vocabulary sizes.** We plot the L2 reconstruction error and the phoneme cluster purity on the out of distribution TIMIT test set.

Since WavCoch is optimized with a single objective–to predict the cochleagram as best as possible–it has no direct incentive to collapse the codebook by merging codes associated with the same phoneme. However, our analyses show that this clustering takes place naturally, to an extent. In future work, we plan to investigate WavCoch models that explicitly incorporate a more efficient codebook, and compare it against the current baseline.

## I.5 Cochleagram vs. Mel Spectrogram

In order to ablate the choice of using the biologically inspired cochleagram representation as our prediction target as opposed to the more standard deep learning practice of using a mel-spectrogram we trained a version of WavCoch using mel-spectrograms as targets. We discovered that the codes have relatively similar vocabulary usage and similar phoneme cluster purity to that learned from cochleagram representation.

We trained models with both representations on the publicly available librispeech960 dataset, consisting of 960 hours of speech recordings. Since the spectrogram L2 reconstruction error is not directly comparable between a cochleagram and mel-spectrogram we utilize two proxy measures: i) Number of unique codes utilized and ii) Phoneme cluster purity. Both of these metrics are computed on the out-of-distribution TIMIT test set. First, related to the number of unique codes utilized: We find that the WavCoch model trained with mel spectrograms utilized 8151 out of 8,192 codes, while the cochleagram version utilized 8,172 codes. Second, related to phoneme cluster purity: We find that the mel spectrogram model achieved an average phoneme cluster purity of 0.3473 while the model trained with the cochleagram achieved an average phoneme cluster purity of 0.3517. While these results are preliminary, they suggest that the cochleagram representation performs at least as well as, if not slightly better than, the mel spectrogram in this context.

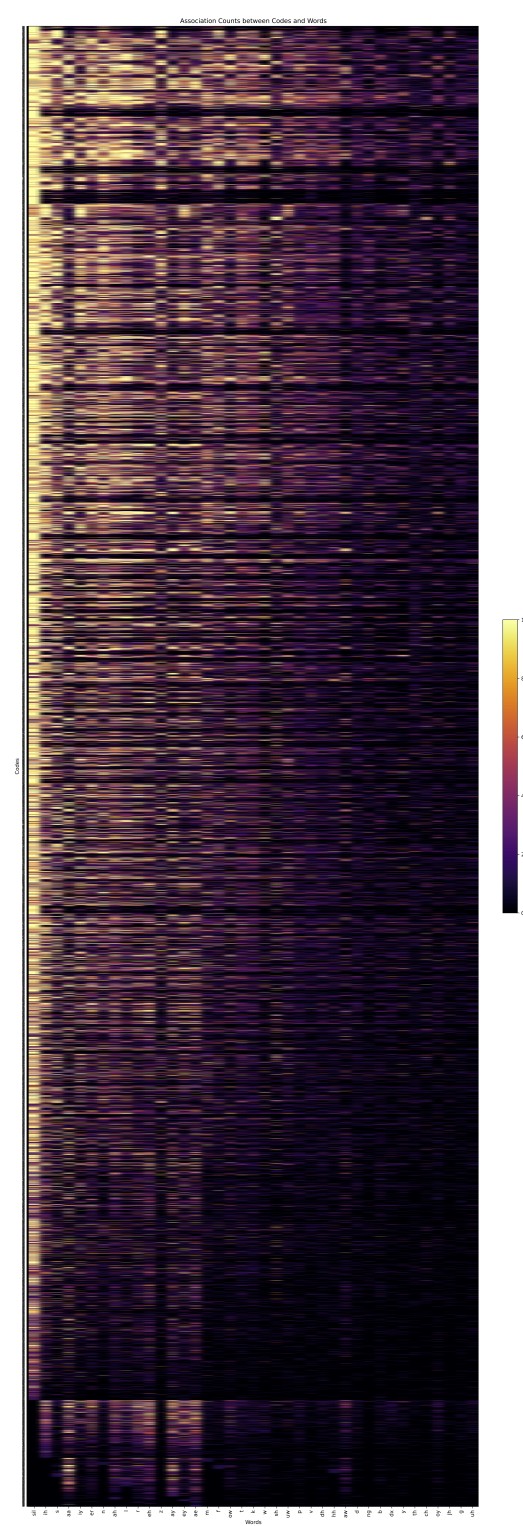

Figure 4: **WavCoch token distribution across phonemes.** The plot indicates token occurrence frequency, correlated with the corresponding phoneme label, on the out of distribution TIMIT test set.

Besides the quantitative analyses reported above, we prefer the cochleagram over the mel-spectrogram representation for conceptual reasons: The ultimate goal of our framework is to move towards more biologically plausible speech models, and the cochleagram is more aligned with this goal.

## I.6 INTERPRETABILITY OF SPEECH CONTINUATION CAPABILITY

We investigate the decoding of CochStream predictions back into the auditory signal. To this end, we developed a simple procedure for inverting the predicted cochleagrams back into a waveform. This procedure is a per-sample optimization of making the waveform match the cochleagram prediction.

Specifically, we optimize a 1D tensor representing the waveform input to make its cochleagram representation match the cochleagram predicted by WavCoch (via L2 error). We backpropagate through the cochleagram transformation and use the Adam optimizer with a learning rate of 1e-2. Note that this optimization procedure is not a learned vocoder model, but a simple procedure of converting the output of WavCoch, the cochleagrams, into audible sound (conceptually similar to Griffin-Lim algorithm).

We upload several audible samples of speech generations from our CochStream: https://anonymous.4open.science/w/cochstream-project-page-0546/. Please access the page using Google Chrome or Firefox as we have seen some cases of Safari not properly loading these videos.

We observe that on short time-scales the model produces reasonable completions, but the longer the completion, the more the predictions drift away from being plausible. We would like to emphasize that the purpose of CochStream is not to be a language model, but a speech representation model – the fact that it can perform rudimentary language modeling is a serendipitous side effect of the training objective, which points to the fact that understanding speech, and producing language can be thought of as a unified objective. These findings serve as great motivating factors for follow-up work which will attempt to stabilize longer term speech generations, building on top of the foundation laid out in this paper.