# OpenReview forum: "Representing speech through autoregressive prediction of cochlear tokens"
_ICLR.cc/2025/Conference — Submitted to ICLR 2025_

### Official Review · Reviewer_Nhso · 2024-10-27

**Soundness:** 2
**Presentation:** 2
**Contribution:** 2
**Rating:** 6
**Confidence:** 5

**Summary:**

The paper introduces a two-stage framework for speech representation learning, drawing on biologically inspired cochlear representations and autoregressive modeling. The first stage, WavCoch, is a vector-quantized encoder designed to convert an STFT spectrogram into a cochleagram representation via MLP layers, a 14-bit Lookup Free Quantization (LFQ) layer, and convolutional layers. The output of the LFQ referred to as “cochlear tokens,” serves as the input for the second stage. The second stage, CochStream, is a GPT-style autoregressive transformer trained to predict the next cochlear token.

To evaluate the effectiveness of these representations, the authors assess CochStream on tasks such as linear phoneme and word probing, lexical semantic similarity, and several tasks from the SUPERB benchmark. CochStream shows excellent performance in lexical semantic similarity and demonstrates competitive results across other tasks, establishing it as a strong contender among state-of-the-art speech representation learning models.

**Strengths:**

1. **Originality**: The paper takes an interesting approach by introducing a cochlear-inspired representation (cochleagram) for speech modeling, which is relatively uncommon in the field.

2. **Quality**: CochStream demonstrates excellent performance on certain tasks, particularly lexical-semantic similarity, suggesting that the model captures semantic features effectively. Additionally, the inclusion of various benchmarks and comparisons to other models is a positive aspect, as it provides a well-rounded view of how CochStream performs across different types of tasks, even if it doesn’t consistently outperform alternatives.

3. **Clarity**: The paper does a good job of describing the high-level design of the WavCoch and CochStream stages, making it reasonably easy to follow the process from quantization to autoregressive modeling. However, some clarity is lost due to inconsistencies in reported training details, which might affect reproducibility.

4. **Significance**: By aiming for a biologically-inspired design, the paper contributes to the broader conversation on incorporating more natural processing methods in machine learning. If refined, the framework could be a step toward more interpretable speech models and potentially inspire future work in biologically plausible approaches.

**Weaknesses:**

**Biological Inspiration**: While the cochleagram representation mimics some aspects of auditory processing, most other components, including the Lookup-Free Quantization (LFQ) and the Transformer, lack biological basis. This raises questions about the validity of the biological inspiration claim.

**Transformer-APC Comparison**: Since CochStream resembles a Transformer-based APC model but with cochlear tokens as input, a direct comparison with an APC trained on the same dataset would strengthen the claim that cochlear tokens are superior to filterbank features or other standard inputs. Without this, the benefit of cochlear tokens over more conventional features is less evident.

**Effectiveness of Separate vs. Integrated Discretization**: Unlike VQ-APC, which integrates discretization within its structure, CochStream uses a separate WavCoch stage. Yet, no experiments justify the choice of a separate discretization stage over an integrated approach. A direct comparison could validate whether the proposed method offers advantages.

**Unclear Advantage of Cochleagram Representation**: The benefit of using a cochleagram over other perceptually inspired representations, like the mel-spectrogram, is not fully demonstrated. Although the paper claims that cochleagram outputs improve interpretability, this interpretability could also apply to other representations. Moreover, the Transformer architecture remains largely a black box, limiting interpretability beyond the output of the WavCoch model.

**Efficiency Claims Not Verified**: While the paper claims efficiency as a strength of cochlear tokens, there is no quantitative analysis to support this. For instance, CochStream’s tokens are sampled at a higher rate (197Hz) than HuBERT’s (50Hz), and cochleagram computation is more costly than mel-spectrograms, albeit only at training time. These factors suggest that cochlear tokens may not be as efficient as implied.

**Inconsistent and Incomplete Training Details**: Key details about the training datasets and parameters are lacking or inconsistent, which raises concerns about reproducibility.

- No citation or details are provided about the 50,000-hour dataset, leaving ambiguity around its source. Using standard datasets like LibriSpeech or LibriLight for English speech tasks would enhance reproducibility.
- In Section 2.1.3, it’s mentioned that WavCoch was trained on 500 hours while CochStream was trained on 50,000 hours, yet CochStream base is reported as trained on 500 hours in the tables.
- Table 2 states CochStream base has 135 million parameters, which contradicts other descriptions in the paper, leading to further confusion.

**Unfair Comparisons**: There are no baselines that are trained on the same dataset, hence it is difficult to put the results into perspective. It remains unclear whether the performance improvements come from the dataset, the transformer configuration or the input representations.

**Improper References**: Several references are cited from the arxiv versions instead of the original published venues. Also, serveral citations lack any details about conference/journal where they were published.

**Questions:**

- A good ablation study is necessary to validate the claims of the paper. Following are some experiments that could help improve the paper:
  - Mel-spectrogram as intermediate representation instead of cochleagram to validate the choice of input representation
  - Cochleagram-based APC trained on the same dataset.
  - Comparison with VQ-APC

- Why were standard publicly available datasets for English speech not utilized for the experiments?

---

> ### Author Response · Authors · 2024-11-16
> **Response to questions requiring clarification and additional details (Part I)**
>
> We appreciate the reviewer’s acknowledgement of the originality, quality, and significance of the work. We thank the reviewer for their careful review. We provide clarifications to the reviewer’s comments/questions, and are currently implementing analyses suggested by the reviewer (training models on open-source data; verifying the efficiency of cochlear tokens), as clarified below.
>
> **Weakness 1. LFQ and Transformer lack biological basis.**
>
> We fully agree that not every part of our pipeline is supported by biological evidence, however, we also emphasize that the precise architecture underlying human auditory processing is not known, and hence we believe that LFQ and Transformers can serve as powerful stand-ins based on the heuristics detailed below.
>
> **LFQ**: While LFQ itself might not be biologically inspired, we believe that our use of it in WavCoch has a biologically sensible interpretation: WavCoch implements a biological transformation (waveform to cochlear representation) and we contend that the LFQ bottleneck can be viewed as a readout of the discrete firing patterns of the neurons along this (largely unknown) pathway. The representations obtained from LFQ are binary 14-bit vectors which can be interpreted as an on-off neural firing pattern [1] which are then processed by downstream circuitry that is not yet well understood (also, please see a more detailed model diagram at the following link, bottom page, Figure 1: https://anonymous.4open.science/w/cochstream-project-page-0546/).
>
> **Transformers**: We provide three reasons why we believe Transformers might serve as a useful approximation for stages of auditory processing:
>
> i) It is well-established that prediction is a neural objective [2] and hence an autoregressive, causal Transformer is a reasonable candidate hypothesis for prediction-based processing in auditory processing at an algorithmic level, but likely not at the implementation-level [3].
>
> ii) Transformers are incredibly powerful for many biologically-plausible tasks, and we wanted to build a model capable of handling various audio-based tasks that humans perform. In the perfect world, we would love to develop a model that perfectly aligns with neural circuitry at a fine-grained implementation-level and is powerful for tasks, but given that we do not know precisely the implementation-level details of auditory processing yet, we think the Transformer serves as a powerful stand-in.
>
> iii) A large transformer with an autoregressive training objective can be viewed as evolving the  circuitry that a biological brain is born with. Computational models did not undergo evolution in the same way as humans (or other species) did – hence, a large training phase where a generic model architecture is exposed to statistical regularities of the world (in this case, speech) may serve to wire up function-specific architecture similar to what a brain is born with.
>
> Finally, we emphasize–as also mentioned in the discussion of our paper–that *“We by no means claim that CochStream is a perfect biologically-inspired model, but it is a critical step in the right direction.”*.
>
> **Weakness 2. Transformer-APC comparison.**
>
> We agree that a direct comparison between a Transformer-based APC model and CochStream would be useful, but given the large amount of additional experiments the reviewers have asked for and our limited resources we will unfortunately not be able to run all of them. Instead, we have included the following paragraph in the paper discussion to make it apparent that we do not claim a superior input representation but rather an advancement of previous work:
>
> *“WavCoch could operate with a mel spectrogram as the target instead of a cochleagram. Our work does not claim superiority of a cochleagram over a mel spectrogram. Instead, the main novelty from WavCoch lies in encoding one representation (in this case, the waveform) into another representation known to be computed in the auditory processing hierarchy (cochleagram) and “probing” an intermediary representation as discrete tokens. We chose the cochleagram for its biological plausibility. Hence, CochStream resembles the APC model [4], but with a quantized input and a Transformer architecture that enables efficient training and scalability.”*
>
> Nevertheless, as discussed below we will dedicate most of our resources to generating a version of our model trained purely on open-sourced data to improve the strength of several comparisons (see response to Weakness 7).

---

> > ### Author Response · Authors · 2024-11-16
> > **Response to questions requiring clarification and additional details (Part II)**
> >
> > **Weakness 3. Effectiveness of separate vs. integrated discretization.**
> >
> > We believe our method offers conceptual and engineering advantages.
> >
> > *Conceptually*, WavCoch can be viewed as mimicking the human cochlea. Since a core goal of our framework is to ultimately get as close as possible to the true human auditory processing stream, we believe that building a separate stage for the inner ear is critical (and this stage can separately be optimized with additional details based on the cochlear basilar membrane etc., and critically, the downstream CochStream Transformer model can similarly be updated).
> >
> > *From an engineering standpoint*, a separate quantization model, which converts the high dimensional raw audio signal into a much lower dimensional token representation, allows us to train the large sequence model by loading the tokens directly into RAM, significantly improving training efficiency and scalability. With the limited hardware available to us, scaling an online tokenization system would be practically infeasible.
> >
> >
> > **Weakness 4. Unclear advantage of cochleagram representation.**
> >
> > WavCoch could operate with a mel spectrogram as the target instead of a cochleagram. We do not make any claims about the superiority of a cochleagram over a mel spectrogram (we have clarified this in the revised manuscript, please see response to Weakness 2 above). Instead, the main novelty from WavCoch lies in encoding one representation (in this case, the waveform) into another representation known to be computed in the auditory processing hierarchy (cochleagram) and “probing” an intermediary representation as discrete tokens. We chose the cochleagram for its biological plausibility.
> >
> >
> > **Weakness 5. Efficiency claims not verified.**
> >
> > *First*, we want to emphasize the differences between our cochlear tokens and HuBERT tokens: Our cochlear tokens serve as a tokenization of the audio input, with the aim of preserving useful information in the raw waveform. Conversely, HuBERT tokens are derived from the embeddings of a deep layer of the model through K-means clustering with the goal of obtaining time-integrated semantic tokens. Hence, while HuBERT tokens have a lower rate than ours (50hz vs. 197hz) they are not directly comparable. Instead, a K-means clustering of 4 consecutive CochStream embeddings from a deep model layer (not cochlear tokens) could be compared to the HuBERT tokens. We believe that our cochlear tokens are more comparable to neural codec tokens such as SoundStream.
> >
> > *Second*, we are conducting two separate experiments to verify the efficiency of cochlear tokens: i) We are investigating the effects of vocabulary size on L2 prediction error on an OOD test set to ensure that your cochlear tokens are indeed efficient. ii) We are currently training a vocoder to evaluate the various encoders to ensure that we are selecting an optimal compression ratio.
> >
> > *Finally*, we note that in our implementation the cochleagram transformation is not the bottleneck during WavCoch training, rendering any efficiency gains over a mel spectrogram unhelpful in terms of overall efficiency (see also Weakness 4 above).
> >
> > **Weakness 6. Inconsistent and incomplete training details.**
> >
> > We deeply apologize for the inconsistencies in the training details. We have fixed and revised the model size and description across the entire manuscript – the error was due to an earlier model version with a larger vocabulary size (which we managed to decrease down to 16,384 as in the current model, and hence do not use the former model version anymore).
> >
> > To sum up, **WavCoch** was trained on 500 hours of data and the parameter count is 5M. **CochStream-base** was trained on 500 hours (different from WavCoch) and the parameter count is 97M. **CochStream-large** was trained on 50K hours (and the parameter count is 1,300M.
> >
> > Please see the Weakness 7 below for response regarding training data.
> >
> > **Weakness 7. Unfair comparisons.**
> >
> > We plan to release the audio dataset as part of a future work, but the release of the data have been delayed. We acknowledge that this is not an ideal situation and offer a solution to this incompleteness: To provide a more fair baseline and enable full reproducibility, we are currently training our entire pipeline using fully open-source datasets (LibriSpeech). We plan on having preliminary results by the end of this rebuttal period and will provide a full analysis in the final manuscript.

---

> > > ### Author Response · Authors · 2024-11-16
> > > **Response to questions requiring clarification and additional details (Part III -- and final, sorry)**
> > >
> > > **Weakness 8. Improper references.**
> > >
> > > We thank the reviewer for pointing this out, and will definitely fix this in the final manuscript.
> > >
> > > **References**
> > >
> > > [1] Levy, William B., and Robert A. Baxter. "Energy efficient neural codes." Neural computation 8.3 (1996): 531-543.
> > >
> > > [2] Spratling, Michael W. "A review of predictive coding algorithms." Brain and cognition 112 (2017): 92-97.
> > >
> > > [3] Marr, David, and Tomaso Poggio. "From understanding computation to understanding neural circuitry." (1976).
> > >
> > > [4] Chung, Yu-An, and James Glass. "Generative pre-training for speech with autoregressive predictive coding." ICASSP 2020-2020 IEEE International Conference on Acoustics, Speech and Signal Processing (ICASSP). IEEE, 2020.

---

> ### Comment · Reviewer_Nhso · 2024-11-19
> **Response to Author's Clarifications**
>
> Thanks for the detailed clarifications and the effort to improve the experimental evaluations, also the updated diagram gives a much clearer picture of the architecture.
>
> I still have some fundamental questions.
>
> Regarding the "probing an intermediate representation as discrete tokens", isn't it the same as obtaining the discrete codes in any VQ-VAE or Neural Codec model, which converts the input representation to an output representation via a quantized bottleneck?
>
> While it is common for VQ-VAE architectures to use the same input and output representations, there are established precedents where input and output representations differ. For example:
>
> 1. **Spectral Codec [1]:** Employs a spectrogram as input and a waveform as the output representation.
> 2. **CLaM-TTS [2]:** Employs mel spectrogram as output representation in its codec.
>
> Given these examples, the work’s primary distinction appears to be the use of a **cochleagram** as the output representation. This biologically inspired choice is an interesting direction, but it primarily reflects an application-oriented novelty rather than a conceptual one.
>
> That said, I am less concerned with the level of novelty and more interested in the **justification for choosing cochleagrams** over other well-established representations, such as mel spectrograms. While the benchmark results demonstrate that cochleagram tokens perform well, it remains unclear whether they offer specific advantages from a biological standpoint that are unique or particularly beneficial for the tasks addressed in the paper.
>
> **Key Questions for Clarification:**
>
> 1. **Comparative Performance:** Would mel spectrogram tokens yield comparable results if used within the same architecture? If so, why prefer cochleagrams?
> 2. **Unique Capabilities:** Are there specific tasks, scenarios, or metrics where cochleagram tokens enable functionality that mel tokens cannot achieve? For instance:
>    - Do cochleagrams provide better alignment with perceptual quality metrics?
>    - Are they more robust to noise or distortions?
>
> While I understand that many reviewers have already requested additional experiments, I want to emphasize that my intention is not to overburden the authors. However, providing some experimental or theoretical justification would significantly strengthen the impact of this work.
>
> [1] Langman, Ryan, et al. "Spectral Codecs: Spectrogram-Based Audio Codecs for High Quality Speech Synthesis." arXiv preprint arXiv:2406.05298 (2024), https://arxiv.org/abs/2406.05298
>
> [2] Kim, Jaehyeon, Keon Lee, Seungjun Chung, and Jaewoong Cho. "CLam-TTS: Improving Neural Codec Language Model for Zero-Shot Text-to-Speech." The Twelfth International Conference on Learning Representations, 2024, https://openreview.net/forum?id=ofzeypWosV.

---

> > ### Comment · Reviewer_Nhso · 2024-11-26
> > **Official Comment by Reviewer Nhso**
> >
> > Do the authors plan to upload the revised paper before the communication deadline?
> >
> > I’d like to check the revised version, if possible, and the clarifications to my follow-up comments before deciding on updating my score.

---

> > > ### Author Response · Authors · 2024-11-26
> > > **Brief response**
> > >
> > > Thanks so much for checking in! We have implemented the requested analyses and are very excited to share them. To ensure clarify and coherence, we have consolidated everything into one comprehensive response which we will post later today (a large global response and an individual one addressing your recent questions). We are looking forward to sharing these additional analyses demonstrating the robustness of our modeling framework.

---

> > > > ### Author Response · Authors · 2024-11-27
> > > > **Brief Update**
> > > >
> > > > Thanks again for your check in. We wanted to give you a quick update: due to some technical difficulties with uploading our audio generations to the anonymous website we were slightly delayed. We will submit both the response and the updated paper pdf as promised sometime before EOD on the 27th (as is the official extended deadline). We apologize for the delay and once again thank the reviewer for the enthusiasm.

---

> > > ### Author Response · Authors · 2024-11-28
> > > **Additional Clarifications**
> > >
> > > We thank the reviewer for the valuable feedback, and for engaging in the discussion. We provide some additional explanations below:
> > >
> > > **Probing an intermediate representation as discrete tokens**
> > >
> > > We thank the reviewer for providing relevant references for our WavCoch quantizer. These have been included in the revised discussion. We would like to point out a few key distinctions with the two references:
> > > CLaM-TTS does indeed auto-encode a mel-spectrogram as the reviewer mentions. In their Mel-VAE pipeline, the mel conversion is performed as a pre-processing step on the input audio, and thus the actual Mel-VAE objective is an autoencoder with the same target as the input to the neural network.
> > >
> > > Similarly, Spectral Codec computes the mel-spectrogram of the input audio, and uses HiFi-GAN as their decoder, which also utilizes the mel-spectrogram as the prediction objective. In contrast, we utilize only the cochleagram as our target objective, allowing the model to learn how to perform the waveform-to-cochleagram transform by itself. The reasoning behind this approach is that the process of learning to transform the sound into the spectral domain itself might yield useful representations of the underlying audio. Taken together, we still believe that WavCoch is novel in its way of learning to transform one representation into another representation through a discrete quantization bottleneck.
> > >
> > >
> > > **Mel-spectrogram vs. cochleagram**
> > >
> > > In order to investigate this important question we trained a version of WavCoch using mel-spectrograms as targets as discussed in our global response. We discovered that the codes have relatively similar vocabulary usage and similar phoneme cluster purity to that learned from cochleagram representation. If anything, the cochleagram representation was slightly superior to mel-spectrograms, but by a small margin. Furthermore, the strong biological evidence for utilizing the cochleagram backs our decision with respect to producing a more biologically-plausible speech model than competing approaches.
> > > Finally, in the revised paper, we have added a discussion on the mel-spectrogram versus cochleagram issue and explicitly note that we do not claim the superiority of cochleagrams over mel-spectrograms.
> > >
> > >
> > > If you have any further questions that are preventing you from raising our score, please let us know and we will do our best to address them in the extended discussion period.

---

> ### Comment · Reviewer_Nhso · 2024-11-28
> **Response to additional clarifications**
>
> I really appreciate the author's efforts in performing the librilight experiments and the cochleagram inversion.
>
> **Time Critical Modifications before paper update deadline**
> 1.  The ablation of the vocabulary size is valuable, and shows the importance of a good ablation study, and may be more useful if included in the main text.
> 2. The table number and title should appear at the top of the table, I missed that in my previous readings of the paper and just realized it. (Mandatory)
> 3. Might I suggest including the appendix with the main paper?
> 4. Include the audio samples (video) as supplementary material if possible, because there is no guarantee that the anonymous link would persist, so maybe the audio samples and the appendix pdf (if not added to the main paper) could be zipped and uploaded as supplementary material.
> 5. I would be cautious about using the term novel framework because, in my opinion, the framework itself is not novel, a two-stage quantized representation followed by auto-regressive modeling is a fairly well-studied framework in several domains, the novelty lies in the specific output representation, and maybe the model architecture of the encoder-decoder. I would suggest the authors to rephrase.
> 6. Thanks for performing the experiments with mel-spectrograms, I think the results are reasonable. I would suggest putting them in a table instead of inline text, that will improve the readability.
>
> **Other Comments**
> 1. I still did not get a satisfactory answer regarding some unique capabilities that might be enabled by using the biologically inspired representation.
> 2. Regarding ClaM-TTS, I agree that the mel-spectrogram is a pre-processing, and the model itself is an autoencoder. However, I disagree with the author's explanation of the Spectral Codec, from my reading of the paper, their input and out representations are different while the decoder architecture is based on hifi-gan, their encoder takes spectrogram and through the band-wise encoders and vector quantizers and the decoder converts them to the waveform, and trained with spectrogram based losses as well as adversarial loss. Thus, it also transforms one representation to another (spectrogram->waveform). Moreover, even in wavcoch, you extract the twiddle factors to first convert the waveform to time-frequency representation (which is not biologically inspired), so the wavcoch is more like transforming one TF representation to another TF representation. As far as I understood, the twiddle factor extraction is not trainable, hence can be considered a pre-processing step as well.
>
> Overall, I think the paper is in a much better shape now, but there is still room for improvement.
>
> I will increase my score to 5 for now.
> To increase it further, I would still need to be convinced about the potential unique benefits, response to point 2 in Other Comments and as much of the "Time Critical Modifications" to be done as possible in the limited time left.

---

> > ### Author Response · Authors · 2024-11-30
> > **Response to Other Comments**
> >
> > We would also like to provide additional responses to the reviewer’s two questions:
> >
> > **Regarding unique capabilities of biologically inspired representation**
> >
> > The cochleagram acts as a band-filter, aligning the perceptual space of the auditory scene to that utilized by humans. Since many AI systems are designed to interact and cooperate with humans, aligning their perceptual spaces is an inherent advantage.
> >
> > As our WavCoch mel-spectrogram ablations have shown, using the cochleagram target provides a slight performance increase in human speech phoneme clustering (as is now discussed in Appendix I.5). We believe that further study of our models in settings such as the Cocktail Party Problem [1] could reveal more such performance gains.
> >
> > We acknowledge that the mel-spectrogram representation is not drastically different from the cochleagram (and this point is explicitly mentioned in the revised paper discussion on OpenReview), however, it is also an inductive bias that we apply, so we argue that we might as well be applying the most human-like inductive bias available to our auditory models.
> >
> > Separately, the field of NeuroAI is interested in both mapping existing AI models to brain recordings and developing biologically plausible models that can perform standard AI tasks [2, 3]. By following the biological constraints as closely as possible we present an (admittedly imperfect) candidate model for the auditory processing pipeline. We believe that developing such models is critical because it illustrates that it is possible to create an AI model with strong task performance while adhering to known biological constraints. Our framework demonstrates competitive performance with models that are less biologically plausible, and we want to push for models in this domain. Finally, we believe that our model with the cochleagram inductive bias will serve as a useful comparison in future work in mapping the internal model representations to brain recordings and analyze how much of the data they can explain in comparison to other AI models that do not adhere to such constraints.
> >
> > **Regarding WavCoch design vs. Spectral Codec**
> >
> > On the [Spectral Codec release page](https://catalog.ngc.nvidia.com/orgs/nvidia/teams/nemo/models/mel_codec_22khz_fullband_medium) they state: “*This 64M parameter model is trained end-to-end using mel and STFT reconstruction losses, and adversarial training with a multi-period descriminator and multi-scale complex STFT discriminator*”. So, one of the model branches goes from a mel-spectrogram input to a mel-spectrogram output, where regression supervision is applied and back propagated through the entire model, making it a true auto encoder. We acknowledge the reviewer’s point that an intermediate waveform representation is generated, however the supervision signal is still (at least in part) derived from the mel-spectrogram reconstruction error.
> > In contrast, WavCoch strictly transforms one representation (the input waveform) into another (the cochleagram), without ever being supervised on the auto-encoding objective, making the function of the model different. We illustrate this distinction in Figure 5 on the website [https://anonymous.4open.science/w/cochstream-project-page-0546/](https://anonymous.4open.science/w/cochstream-project-page-0546/).
> >
> > We also argue that our design is more biologically plausible since it performs a transformation known to occur in the human auditory processing pipeline.
> >
> > Regarding the twiddle factors, we acknowledge that in the current frozen state they can be viewed as a pre-processing step, however even in that case WavCoch is strictly performing a feature extraction process from the time-frequency feature space, not an auto encoding process, unlike the Spectral Codec (since loss flows only from one representation to another). Furthermore, we implement the twiddle factors as a 1D convolutional layer (we are happy to provide the code for this in our supplementary zip archive if it would help clarify things). We experimented with a variation where the twiddle factors are learned from scratch – that model worked as well but required a bit longer to properly converge. We also experimented with a configuration where the twiddle factors are initialized, and then trained – this made very little difference. Perhaps we should treat the twiddle factors as a special initialization of a 1D convolution in a future revision to minimize the confusion. Finally, these twiddle factors can be thought of as mimicking the functionality of the cochlea - a structure that is evolved and not learned in biological systems which performs frequency separation through the physics of the geometry [4]. We thus argue that initializing the twiddle factors is a reasonable stand in for the function of this structure.

---

> > > ### Author Response · Authors · 2024-11-30
> > > **References for Response to Other Comments**
> > >
> > > **References:**
> > >
> > > [1] Bronkhorst AW. The cocktail-party problem revisited: early processing and selection of multi-talker speech. Atten Percept Psychophys. 2015 Jul;77(5):1465-87. doi: 10.3758/s13414-015-0882-9. PMID: 25828463; PMCID: PMC4469089.
> > >
> > > [2] Kell, A. J., Yamins, D. L., Shook, E. N., Norman-Haignere, S. V., & McDermott, J. H. (2018). A task-optimized neural network replicates human auditory behavior, predicts brain responses, and reveals a cortical processing hierarchy. Neuron, 98(3), 630-644.
> > >
> > > [3] Millet, J., Caucheteux, C., Boubenec, Y., Gramfort, A., Dunbar, E., Pallier, C., & King, J. R. (2022). Toward a realistic model of speech processing in the brain with self-supervised learning. Advances in Neural Information Processing Systems, 35, 33428-33443.
> > >
> > > [4] Casale J, Kandle PF, Murray IV, et al. Physiology, Cochlear Function. [Updated 2023 Apr 1]. In: StatPearls [Internet]. Treasure Island (FL): StatPearls Publishing; 2024 Jan-. Available from: https://www.ncbi.nlm.nih.gov/books/NBK531483/

---

> > > > ### Comment · Reviewer_Nhso · 2024-12-01
> > > > **Response**
> > > >
> > > > Thanks for the explanations.
> > > >
> > > > I am convinced about the capabilities of the biologically inspired representation.
> > > >
> > > > However, I still do not feel too comfortable with the novelty claim of the "transform one representation to another".
> > > > I feel that the primary scientific contribution (and I believe an important one) is the application and analysis of a biological representation under a modern speech representation learning framework of speech tokenization + AR modeling, and showing competitive performance in several tasks demonstrating the plausibility of those features for speech representation learning and ML tasks.
> > > >
> > > > Finally, I will not look at the modified paper in the anonymous website, since it was submitted after the deadline, I will consider the version currently on openreview. While this may seem harsh, I think it is fair since the deadline was same for everybody. This was also one of the reasons why I asked for the updated paper with a couple of days to go, so that any final modifications could be finished in time.
> > > >
> > > > I will keep the current score for now.

---

> > > > > ### Author Response · Authors · 2024-12-01
> > > > >
> > > > > We really appreciate your rapid response and engagement.
> > > > >
> > > > > We appreciate your comment regarding the importance of biologically-plausible, task-performant models, and agree with you that this is the core contribution of the paper and should be highlighted as such.
> > > > >
> > > > > We want to clarify that we do not seek to make novelty claims regarding the general architecture of our WavCoch quantizer (and all such claims have been removed in our recently revised manuscript version). We specifically want to highlight the idea that biologically inspired systems could use a similar approach to quantize representations. This general idea is crucial because most tokenization schemes rely on some sort of auto-encoding – a process which to the best of current understanding appears biologically implausible. Even speech produced in response to an auditory prompt (the closest analogue to an auto-encoder) very poorly reconstructs the input sound according to some distance metric such as L2 applied to the predicted waveform/spectrogram. In contrast, there are several well-studied transformations which occur either subcortically or in the brain , providing biologically sound targets for a quantization model which mimics the function of that specific part of a processing hierarchy. If we re-scope our claims regarding the novelty of our approach to focus specifically on biologically inspired NeuroAI models, would that make you more comfortable with our claim? Specifically, we will remove all claims about the novelty of transforming one representation into another, and instead focus on the fact that this type of transformation-to-transformation design is critical in models that aim to align with biological processes. Please let us know if you prefer another way forward.
> > > > >
> > > > > We absolutely acknowledge that the deadline for a new upload has passed, and completely understand your stance. Given the change of deadlines for the discussion period and your request for additional modifications we were slightly unsure about whether we would be able to update the PDF again, so we tried our best, but this is certainly our fault. We nevertheless appreciate you making an effort to allow us to fix those details.
> > > > >
> > > > > Finally, regarding the “Modifications” in your [former response](https://openreview.net/forum?id=TQdg1X6eqm&noteId=5E7lOO3MqW), we absolutely agree with all of your suggestions. Besides the novelty claim of WavCoch (as clarified in above), we believe your comments pertain to cosmetic and minor structural points, and if the paper gets accepted we will be sure to include them all in the camera-ready version (as they make the paper much more readable! Thanks again).

---

> > > > > > ### Comment · Reviewer_Nhso · 2024-12-02
> > > > > > **Response**
> > > > > >
> > > > > > Thanks for the clarifications and commitment to improve the paper.
> > > > > > > This general idea is crucial because most tokenization schemes rely on some sort of auto-encoding – a process which to the best of current understanding appears biologically implausible. Even speech produced in response to an auditory prompt (the closest analogue to an auto-encoder) very poorly reconstructs the input sound according to some distance metric such as L2 applied to the predicted waveform/spectrogram. In contrast, there are several well-studied transformations which occur either subcortically or in the brain , providing biologically sound targets for a quantization model which mimics the function of that specific part of a processing hierarchy. If we re-scope our claims regarding the novelty of our approach to focus specifically on biologically inspired NeuroAI models, would that make you more comfortable with our claim? Specifically, we will remove all claims about the novelty of transforming one representation into another, and instead focus on the fact that this type of transformation-to-transformation design is critical in models that aim to align with biological processes.
> > > > > >
> > > > > > I think this is a good way to summarize and state the motivation and direction of the research, and if added would definitely improve the appeal of the paper.
> > > > > >
> > > > > > I will raise my score to 6.

---

> ### Author Response · Authors · 2024-11-29
> **Time Critical Modifications**
>
> Thank you so much for your continued engagement and valuable feedback.
>
> We have implemented all of the time critical fixes to the paper itself as you suggested:
> - We moved a short description of the WavCoch vocabulary size ablation to the main text and pointed to the Appendix for additional discussion and figure (for space constraint reasons).
> - We fixed all the tables to conform to the format of caption above the body (thank you for catching this!)
> - We merged the appendix into the main paper PDF and linked all the references.
> - We created a zip file of all the sample videos and a README document and turned it into supplementary material. We plan on releasing a website for our project, that will have even more samples and will be hosted on GitHub (and thus be persistent), but it is nevertheless good to have a backup.
> - We have modified the text to no longer refer to the two stage framework as novel, and have re-phrased all the places in which we claimed "novelty" of the framework. Instead, we explicitly refer to it as a "two-stage framework".
> - We put the Mel spectrogram ablation results in a table.
>
> After completing the above fixes in less than 24 hours after receiving your comment, we found that we are unable to upload another version of our PDF and supplementary material. We sincerely apologize for this. In order to provide a temporary remedy we uploaded a version of the paper with all the edits described above to our anonymous GitHub page [here](https://anonymous.4open.science/w/cochstream-project-page-0546/assets/files/11854.pdf), as well as the supplementary [zip file](https://anonymous.4open.science/w/cochstream-project-page-0546/assets/files/11854-supp.zip).
>
> We once again truly apologize for this and promise to upload the fixed version for the camera ready submission upon acceptance.

---

### Official Review · Reviewer_6c7S · 2024-10-30

**Soundness:** 3
**Presentation:** 3
**Contribution:** 3
**Rating:** 6
**Confidence:** 5

**Summary:**

The paper proposes a self-supervised learning (SSL) framework to learn generic speech representations using cochlear tokens derived from a biologically inspired waveform-to-cochleagram transformation, referred to as WavCoch. These cochlear tokens serve as input representations for an auto-regressive model, CochStream, which aims to emulate the human auditory processing stream.

**Strengths:**

The paper is well-written, with a clear and interesting motivation rooted in mimicking the human auditory system. The authors’ use of cochleograms as intermediate acoustic representations is compelling, and the performance of these biologically inspired features on downstream tasks is promising. The results show that hand-crafted, biologically motivated features can indeed achieve competitive performance, which could have significant implications for the design of speech processing systems.

**Weaknesses:**

While the paper achieves competitive performance on benchmark tasks, it relies on general-purpose, objective ML measures to validate the proposed features. This focus shifts away from the original biological motivation to a more standard ML performance evaluation. For researchers focused on ML, data-driven features generally remain more attractive due to better performance across tasks and the lack of a need for hand-crafting input features. To realign with the biological motivation, the paper would benefit from additional biologically relevant evaluations, such as subjective metrics from human listening tests, to better demonstrate the utility of these handcrafted features for tasks where biological relevance is key.

**Questions:**

Performance on Non-Content Tasks:
Given that WavCoch is designed based on human auditory processing, it would be expected to excel in tasks that are less content-focused. The promising emotion recognition results align with this expectation, but the lower performance on speaker identification is less encouraging. Could the authors consider discussing this disparity to clarify the strengths and limitations of the cochleogram-based representations for non-content-focused tasks.

Speech Continuation Capability:
In Section 3.3, the authors include examples to illustrate CochStream’s speech continuation ability. However, assessing this capability solely by visual inspection of cochleograms is challenging without quantitative metrics or synthesized speech outputs. As the framework is intended to mimic human auditory processing, could the authors incorporate a subjective evaluation from human annotators on synthesized outputs to strengthen the claims of biologically motivated capabilities?

Motivation for Auto-Regressive Training:
The paper currently lacks clarity on why an auto-regressive model is preferred over, for instance, a BERT-style masking training approach. Is the primary intent to enable speech continuation? This could be clarified. Providing further justification here would add clarity to the design decisions.

---

> ### Author Response · Authors · 2024-11-16
> **Response to questions requiring clarification**
>
> We are very encouraged by the reviewer’s positive feedback and acknowledgement of the impact of the work, and are thankful for their thoughtful comments. Below, we clarify the reviewer's questions, and will provide an updated response when the suggested vocoder has been implemented.
>
> **Weakness. Reliance on ML benchmarks.**
>
> *First*, we appreciate the reviewer highlighting the importance of biologically-relevant benchmarking, which we agree demonstrates a significant gap in the literature. While ML benchmarks are of course standard for evaluating audio-based models, the lack of established biologically-relevant benchmarks becomes evident as (a subset of) the field moves toward more human-inspired models. Through this work, and in planned follow-up studies, we aim to contribute to closing this gap.
> We want to emphasize that we included the lexical similarity benchmark (sSIMI), which is not a fully standard ML benchmark as it involves direct comparison to human performance. Hence, this represents an initial step toward incorporating more human-based evaluations.
>
> *Second*, as the reviewer mentions, we believe that a fundamental advantage of CochStream is that it can naturally be prompted with sounds (much like a language model can be prompted with text). This capability can indeed pave the way for a new era of cognitively inspired prompt-benchmarking of auditory models. Since our model is unique in this capability we were unable to compare it to any baselines, however we will attempt to perform a pilot evaluation of the continuations by training a vocoder to obtain audible continuations (also see response to Question 2 below) during the rebuttal period. We will include a detailed discussion of this paradigm and its implications in the final manuscript.
>
> *Third*, if the reviewer has a particular biologically-inspired evaluation in mind for the current paper, we would be happy to test our model on it (depending on the benchmark’s scope, of course). Otherwise, we are very interested in pursuing this direction in future papers, once we have introduced the model in the current paper.
>
> **Question 1. Performance on non-content tasks.**
>
> We will provide a more thorough analysis of our model’s performance characteristics and unique advantages by the end of the rebuttal period, including addressing the disparity between the performance on different types of tasks.
>
> **Question 2. Speech continuation capability.**
>
> To perform some preliminary analysis of our model’s speech continuation capability, we are currently training a vocoder model to decode the continuations back to audio. We will show some preliminary results by the end of the rebuttal period and will include a deeper analysis, including qualitative human evaluations, in the final manuscript.
>
> We also want to emphasize that the focus of the current paper is *speech representation learning*, and not *language modeling* and hence we leave the longer continuations and linguistic benchmarks for a separate paper.
>
> **Question 3. Motivation for autoregressive training.**
>
> We chose the autoregressive training objective for three main reasons (which we will include in the discussion of the revised manuscript):
>
> i) Amongst the algorithms commonly used in modern AI, we believe autoregressive learning is the most biologically plausible one. Unlike the BERT objective, it does not violate causality, and does not require us to discover units through global k-means clustering for subsequent use in autoregressive sequence models. Unlike contrastive objectives it does not require a large selection of heuristically-picked positive and negative examples. Instead it simply predicts the distribution of next plausible sounds.
>
> ii) Autoregressive training allows for the evaluation of the surprisal associated with any given sequence of sounds, and for the speech continuation capability discussed in Question 2 above.
>
> iii) Autoregressive training is simple, yet capable of capturing enormous complexity when appropriately scaled. In the paper, we demonstrate that the autoregressive objective scales well within the speech domain (comparison of CochStream-base versus CochStream-large).

---

> ### Author Response · Authors · 2024-11-28
>
> Thanks again for the time spent reviewing our work. We have now addressed all of your questions and concerns through the individual response (posted last week) as well as through the global response. If you have any further questions that are preventing you from raising our score, please let us know and we will do our best to address them in the extended discussion period.

---

> ### Author Response · Authors · 2024-12-02
> **Approaching Deadline**
>
> We would once again like to thank the reviewer for their comments and suggestions. We have addressed each one of the concerns raised by the reviewer either through a text response or with follow up experiments.
>
> Specifically, we:
> - Developed a procedure for inverting the predicted cochleagrams into audio, and generated several rollout samples available at the bottom of out anonymous website (https://anonymous.4open.science/w/cochstream-project-page-0546/). We really hope the reviewer will take a look at these, as they provide great intuition about the model’s predictions and the interpretability advantage of our framework.
> - Responded with clarifications behind our reasoning for choosing the autoregressive training objective. We also wanted to offer another clarification on task-performance: Some of the SUPERB task performance patterns, where the cochleagram representation did not yield the expected performance characteristics, can likely be attributed to the naturalistic dataset used to train our initial model. To address this, we trained a new version of the model on the Libri-Light dataset, which is also used by the baseline comparison models. Although we could only evaluate the halfway checkpoint due to time constraints, this version demonstrated strong performance compared to the model trained on our original dataset. We anticipate that the fully trained model will outperform the initial version on many of the SUPERB tasks. As a final point, we note that the Libri-Light dataset, comprising clean recordings of English audiobooks with single-speaker consistency throughout, offers characteristics particularly well-suited for pre-training certain tasks. Previous research has shown that the choice of training data can be more critical than model architecture or task objectives in shaping model representations [1]. In the camera-ready version of this paper, we plan to present a comprehensive ablation study examining how different datasets influence task performance. Additionally, we will explore a “best of both worlds” approach by training a model on a combination of our naturalistic data and the Libri-Light dataset.
>
> Since it is getting close to the end of the rebuttal period and we believe that we have addressed all of the stated weaknesses we kindly ask the reviewer to let us know if there are any further unaddressed concerns–otherwise we would be grateful if you could adjust the score accordingly.
>
> **References:**
> [1] Conwell, C., Prince, J.S., Kay, K.N. et al. A large-scale examination of inductive biases shaping high-level visual representation in brains and machines. Nat Commun 15, 9383 (2024). https://doi.org/10.1038/s41467-024-53147-y

---

### Official Review · Reviewer_npk7 · 2024-11-02

**Soundness:** 3
**Presentation:** 3
**Contribution:** 2
**Rating:** 5
**Confidence:** 4

**Summary:**

This paper introduces CochStream, a biologically-inspired model for representing speech by predicting sequences based on auditory representations akin to the human cochlea. The model operates in two stages. (1) WavCoch Encoding: The first stage transforms raw audio into a time-frequency cochleagram (a cochlea-inspired representation), effectively converting the continuous audio into discrete "cochlear tokens." (2) Autoregressive Prediction with CochStream: Using these tokens, CochStream—an autoregressive Transformer—predicts the next token in the sequence, creating a generative model that can continue audio sequences and interpret speech patterns.

**Strengths:**

- CochStream explores an audio representation generation method inspired by the human cochlea.
- CochStream was evaluated on the SUPERB benchmark for tasks such as speech recognition, intent classification, and speech separation.
- The model can visualize predictions in cochleagram form, offering interpretable insights into its speech representations.

**Weaknesses:**

- The baseline model seems a bit weak and should try to incorporate newer and more powerful models such as Whisper; additionally, CochStream claims to have the appearance of acoustic information, so that should be compared with models that aim at audio reconstruction such as Encodec, DAC, Soundstream;
- Based on the results in Tables 1 and 3, the performance improvement of CochStream over the baseline models appears limited. I would like the authors to further clarify the main advantages of CochStream compared to traditional approaches.
- The design of CochStream reminds me of recent popular single-codebook audio reconstruction approaches, such as Single-Codec and WavTokenizer. I noticed that CochStream has a vocabulary size of 16,384. Have the authors explored the impact of vocabulary size on performance and the vocabulary utilization rate? Is there any occurrence of collapse?
- CochStream seems to complete wav continuation based on a seed. I’m curious whether it has the ability to generate long-sequence audio and how effective it is.
- In fact, CochStream aims to explore audio representation from a biological perspective, but I believe that this simple autoregressive modeling approach cannot adequately simulate the complex processing mechanisms of biological systems.

**Questions:**

See details in Weaknesses.

---

> ### Author Response · Authors · 2024-11-15
> **Experiments on vocabulary utilization rate and response to all questions requiring clarification (Part I)**
>
> We thank the reviewer for their positive feedback and insightful questions! Below we describe experiments investigating vocabulary utilization rate and answer questions requiring conceptual clarification.
>
> **Weakness 1. Baseline models seem weak.**
>
> We will include an audio autoencoding (neural codec) model baseline as the reviewer suggested. We believe that Whisper is not a fair comparison given that it is trained on labeled text data. Instead, we believe that other self-supervised models are fair comparisons.
>
> **Weakness 2. Main advantages of CochStream.**
>
> We are working on including additional comparison models per the reviewers’ comments, and following these results, we will provide a more thorough analysis of our model’s advantages by the end of the rebuttal period.
>
> **Weakness 3. Vocabulary utilization rate of WavCoch.**
>
> We thank the reviewer for raising this point. We would like to clarify that the cochlear tokens are meant to be a lightweight encoding of the underlying auditory stream (much like BPE tokens for text), not a deep semantic clustering of features. Nevertheless, we conducted some cluster purity experiments and surprisingly found that even at the token level the model starts to extract structures that are loosely related to phonemes.
>
> We compiled some initial results into a plot which can be found at the bottom of our anonymous website (https://anonymous.4open.science/w/cochstream-project-page-0546/) as Figure 2. First of all, we asked how many tokens were utilized for the out of distribution TIMIT test set. We found that 82.5% tokens were utilized (13,517 tokens out of 16,384).
>
> Second, we investigated the vocabulary utilization rate in relation to the phoneme purity of each token. For each token, we define the purity as:
>
> Purity = (Count of most associated phoneme for token i) / (total counts for token i)
>
> Hence, the purity (on Figure 2’s y-axis) shows the purity for each token, whereas the color shows how many times a given token appears in the test set. As evident from the plot, even tokens that occur more than 70 times each have high purity values, up to 0.85.
>
> Inspired by these initial results we will conduct a more thorough investigation of the cochlear token space, with varying bottleneck sizes during the course of the rebuttal period. We will include a detailed section on these results in the revised manuscript.
>
> **Weakness 4. Long continuations based on a seed.**
>
> It is correctly observed that CochStream generates relatively long waveform continuations that appear realistic. To perform some preliminary analysis of their quality, we are training a vocoder model to decode the continuations back to audio. We will show some preliminary results by the end of the rebuttal period and will include a deeper analysis in the final manuscript.
>
> We also want to emphasize that the focus of the current paper is *speech representation learning*, and not *language modeling* and hence we leave the longer continuations and proper linguistic evaluations for a separate paper.

---

> ### Author Response · Authors · 2024-11-15
> **Experiments on vocabulary utilization rate and response to all questions requiring clarification (Part II)**
>
> **Weakness 5. Autoregressive modeling does not capture the complexity of biological systems.**
>
> We agree with the reviewer’s point: it is likely that an algorithm that captures the full complexity of biological systems might need to include different kinds of objectives, recurrence, wiring constraints, different cell types etc. – that said, very little is known about the neural architecture and learning rules underlying auditory processing in humans, and hence we believe that we cannot a priori reject autoregressive learning. Below, we emphasize three points as to why autoregressive Transformers might serve as a useful approximation in the current context:
>
> i) It is well-established that prediction is a neural objective [1] and hence an autoregressive Transformer is a reasonable candidate hypothesis for prediction-based processing in auditory processing at an algorithmic level, but likely not at the implementation-level [2].
>
> ii) Transformers are incredibly powerful for many biologically-plausible tasks, and we wanted to build a model capable of handling various audio-based tasks that humans perform. In the perfect world, we would love to develop a model that perfectly aligns with neural circuitry at a fine-grained implementation-level and is powerful for tasks, but given that we do not know precisely the implementation-level details of auditory processing yet, we think the Transformer serves as a powerful stand-in.
>
> iii) A large transformer with an autoregressive training objective can be viewed as evolving the  circuitry that a biological brain is born with. Computational models did not undergo evolution in the same way as humans (or other species) did – hence, a large training phase where a generic model architecture is exposed to statistical regularities of the world (in this case, speech) may serve to wire up function-specific architecture similar to what a brain is born with.
>
> Finally, as we emphasize in the discussion of the paper, *“We by no means claim that CochStream is a perfect biologically-inspired model, but it is a critical step in the right direction.”*. We believe our paper is a big step in the right direction through i) more biologically plausible input representations compared to other quantization schemes, ii) no global clustering procedures, iii) no large intra-batch comparisons, and iv) no bidirectionality. In the discussion, we additionally refer to several works that argue that Transformer computations can be implemented in biological tissue [3-5].
>
>
> **References** \
> [1] Spratling, Michael W. "A review of predictive coding algorithms." Brain and cognition 112 (2017): 92-97. \
> [2] Marr, David, and Tomaso Poggio. "From understanding computation to understanding neural circuitry." (1976). \
> [3] Bricken, T., & Pehlevan, C. (2021). Attention approximates sparse distributed memory. Advances in Neural Information Processing Systems, 34, 15301-15315. \
> [4] Whittington, James CR, Joseph Warren, and Timothy EJ Behrens. "Relating transformers to models and neural representations of the hippocampal formation." arXiv preprint arXiv:2112.04035 (2021). \
> [5] Kozachkov, Leo, Ksenia V. Kastanenka, and Dmitry Krotov. "Building transformers from neurons and astrocytes." Proceedings of the National Academy of Sciences 120.34 (2023): e2219150120.

---

> ### Author Response · Authors · 2024-11-28
>
> Thanks again for the time spent reviewing our work. We have now addressed all of your questions and concerns through the individual response (posted last week) as well as through the global response. If you have any further questions that are preventing you from raising our score, please let us know and we will do our best to address them in the extended discussion period.

---

> ### Author Response · Authors · 2024-12-01
> **Approaching Deadline**
>
> We would once again like to thank the reviewer for their comments and suggestions. We have addressed each one of the concerns raised by the reviewer either through a text response or with follow up experiments.
>
> Specifically, we:
> - Added additional baselines to our revised results tables (see Tables 1 and 2 in the revised paper).
> - Added additional information regarding the advantages of the cochlear representation and our model more broadly, as well as comparisons to more recent approaches in the field (see Section 4).
> - Ran ablation experiments to discover the optimal vocabulary size for our model (see Appendix Section I.3) This actually resulted in us finding an improved vocabulary size for our tokenization model (one bit smaller than the previous one) – thank you!
> - Developed a procedure for inverting the predicted cochleagrams into audio, and generated several rollout samples available at the bottom of out anonymous website (https://anonymous.4open.science/w/cochstream-project-page-0546/). We really hope the reviewer will take a look at these, as they provide great intuition about the model’s predictions and the interpretability advantage of our framework.
> - Wrote a lengthy reply to questions regarding to what extent the simple autoregressive modeling approach can capture the complexity of biological systems in our initial reply (https://openreview.net/forum?id=TQdg1X6eqm&noteId=MOlIO5Qbhe) and made modifications to the discussion section of the paper (see Section 4).
>
> Since it is getting close to the end of the rebuttal period and we believe that we have addressed all of the stated weaknesses we kindly ask the reviewer to let us know if there are any further unaddressed concerns–otherwise we would be grateful if you could adjust the score accordingly.

---

> > ### Comment · Reviewer_npk7 · 2024-12-02
> >
> > I appreciate the author's clarification, but I have not found any experimental results comparing with the neural codec model. Is there any additional information on this?

---

> > > ### Author Response · Authors · 2024-12-03
> > > **Clarification About Neural Codec Experiments**
> > >
> > > Thank you for the engagement and the reminder.
> > >
> > > We apologize for not clarifying this earlier: we conducted a probing experiment on several neural audio codec models but found their performance to be poor. This result is not unexpected, as the codecs are primarily trained with a reconstruction objective focused on efficiently compressing information rather than extracting robust representations for downstream tasks. Similarly, the embeddings of our quantizer, WavCoch, perform poorly compared to those of our downstream sequence prediction model, CochStream.
> > >
> > > Specifically, we probed the official implementation of the [Encodec model](https://github.com/facebookresearch/encodec) and an open-source implementation of the [SoundStream model](https://github.com/haydenshively/SoundStream/tree/fb850e541add09f31634344d6c7113d65a2aadfc). We evaluated these models on word and phoneme probing tasks as well as the sSIMI semantic similarity task. After conducting an extensive search across layers and pooling methods, we determined that Max pooling yielded the best results for both codecs on all tasks. Furthermore, we found that the layer immediately preceding quantization contained the most effective representations. In general Encodec performed stronger than SoundStream. The results are summarized in the tables below:
> > >
> > >
> > > **Table 1: Linear probing performance for phonemes or words on the TIMIT dataset**
> > >
> > > | **Model Name**             | **Phoneme Decoding** | **Phoneme Random** | **Word Decoding** | **Word Random** |
> > > |--------------------------|-----------------------|--------------------|-------------------|-----------------|
> > > | **HuBERT-xl**            | **0.93**             | 0.20              | **0.88**          | 0.07           |
> > > | **HuBERT-base**          | 0.85                 | 0.20              | 0.77              | 0.07           |
> > > | **wav2vec2-large**       | 0.79                 | 0.20              | 0.43              | 0.07           |
> > > | **WavLM-large**          | 0.91                 | 0.20              | 0.85              | 0.07           |
> > > | **CochStream-base**      | 0.82                 | 0.20              | 0.48              | 0.07           |
> > > | **CochStream-large**     | 0.92                 | 0.20              | 0.67              | 0.07           |
> > > | **CochStream-large-ll**  | 0.92                 | 0.20              | 0.69              | 0.07           |
> > > | **Encodec Full**         | 0.61                 | 0.20                 | 0.23              | 0.07              |
> > > | **Soundstream Full**     | 0.30                 | 0.20                 | 0.11              | 0.07              |
> > >
> > >
> > >
> > >
> > > **Table 2: Semantic similarity scores on the ZeroSpeech 2021 Lexical Semantic Benchmark**
> > >
> > > | Dataset               | LibriSpeech Audio Accuracy ↑ | Synthetic Audio Accuracy ↑ |
> > > |-----------------------|------------------------------|-----------------------------|
> > > | HuBERT-xl            | 7.81                        | 10.37                       |
> > > | HuBERT-base          | 6.10                        | 7.48                        |
> > > | wav2vec2-large       | 6.41                        | 7.19                        |
> > > | WavLM-large          | 10.50                       | 10.41                       |
> > > | CochStream-base      | 10.63                       | 10.12                       |
> > > | CochStream-large     | **12.52**                   | **10.64**                   |
> > > | CochStream-large-ll  | 10.99                       | 10.52                       |
> > > | Encodec sSSIMI       | 10.07                       | 8.00                        |
> > > | Soundstream sSSIMI   | 9.51                        | 7.29                        |
> > >
> > >
> > > Since these are different classes of models that behave significantly differently from the other models evaluated (all of which are speech representation models), we initially excluded them from the main results tables.
> > >
> > > However, following your note and in light of the surprisingly strong performance of these neural codecs on the sSIMI task, we have decided to include them in the main tables with a brief discussion of their performance in the camera-ready version.
> > >
> > > Please let us know if you have any further questions.

---

### Official Review · Reviewer_ipew · 2024-11-06

**Soundness:** 2
**Presentation:** 2
**Contribution:** 2
**Rating:** 5
**Confidence:** 4

**Summary:**

This paper presents CochStream, a biologically-inspired two-stage framework for speech representation learning. The model leverages human cochlear-inspired audio processing to create a discrete "cochlear token" representation, which it then feeds into an autoregressive Transformer model to predict future tokens. This approach differs from traditional signal-reconstruction and contrastive models, aiming instead for a representation that captures the hierarchical structure of human auditory processing. Experimental results show that CochStream performs competitively on phoneme recognition, word decoding, and lexical semantics tasks, often surpassing existing baselines on the SUPERB benchmark.

**Strengths:**

1.	The model's biologically-inspired cochlear token framework is well-aligned with human auditory processing, making it a promising approach for more human-like and interpretable speech representations.

2.	The authors validate the model’s versatility across various tasks (phoneme/word decoding, lexical semantics) and benchmarks (e.g., SUPERB), demonstrating its competitive performance and interpretability advantages over existing models.

**Weaknesses:**

1.	The paper lacks formulas and a clear explanation of the cochlear representation, as well as a model architecture diagram. This makes it difficult to understand how the cochlear representation is converted into audio and how it compares to or provides advantages over the mel representation. A more thorough theoretical or visual explanation of the cochlear encoding process would enhance clarity.


2.	The experimental results on linear probing performance for phonemes and words on the TIMIT dataset are insufficient. Notably, CochStream-base is not compared with similarly parameterized models such as WavLM-base or wav2vec2-base, which limits the ability to assess CochStream's true effectiveness relative to models of similar scale. Additionally, CochStream-large demonstrates only average performance in Word Decoding. The model's embedding performance on various downstream tasks in the SUPERB benchmark also does not clearly showcase a significant advantage over existing models.

**Questions:**

1.	Why is section 2.1.4 OBTAINING COCHSTREAM EMBEDDINGS empty?

2.	Due to the lack of information in OBTAINING COCHSTREAM EMBEDDINGS, I am unsure how the representation predicted by this GPT-style autoregressive Transformer compares in speed to representations from models like HuBERT, which use masked prediction. Could the authors clarify this aspect?

---

> ### Author Response · Authors · 2024-11-15
> **Response to all questions requiring clarification or additional information.**
>
> We thank the reviewer for their careful review and thoughtful feedback! Below, we address all the reviewer’s questions (in two separate comments), except for the comparison to additional baseline models and a more thorough analysis of model performance, which we are currently working on. We are happy to further clarify any of the answers below.
> \
> \
> **Weakness 1. Lacking clear explanations of the cochlear encoding process, its relation to mel spectrograms, and model diagrams.**
>
> We have created a detailed model diagram, demonstrating the WavCoch architecture as well as the CochStream architecture, please see Figure 1 uploaded at this anonymous link (bottom of the page): https://anonymous.4open.science/w/cochstream-project-page-0546/.
>
> We will include the model diagram in the revised manuscript along with a detailed description:
>
> **WavCoch Architecture**: First, the raw waveform (shape: 1,80000 for 5s of mono audio sampled at 16kHz) undergoes the Fourier Transform by computing Twiddle Factors [1]. These factors represent complex sinusoidal components that decompose the signal into its frequency spectrum. The Twiddle Factors are applied to the audio signal through a 1D convolution (window size 1,001 and hop length 80 samples) which transforms the signal into the time-frequency domain. \
> Second, each 5 ms temporal step of this time-frequency representation is fed into two fully-connected (FC) layers with ReLU nonlinearities (with 512 hidden units each). \
> Third, these embeddings are then passed through a 14-dimensional LFQ bottleneck [2], which effectively binarizes the representation. We read out the activations of this bottleneck as a 14-bit binary code which can be interpreted as one of 2^14 = 16,384 discrete tokens. \
> Fourth, the output of the LFQ bottleneck is then projected to a 211 dimensional output, through two 1-dimensional convolutional layers (kernel size 10 and stride 1), separated by ReLU nonlinearities. This output corresponds to the frequencies in the cochleagram representation [3] which it is supervised to match via L2 error.
> Thus for every 5 seconds of audio, WavCoch extracts a sequence of 988 integers in the range [0, 16384) through the LFQ bottleneck, denoted as **cochlear tokens**, to feed into CochStream.
>
> **CochStream Autoregressive Model Architecture**: The cochlear tokens obtained in WavCoch are passed to a GPT-style autoregressive Transformer [4], denoted as CochStream. We train two versions: CochStream-base (97M parameters), with 12 layers, 12 attention heads and an embedding size of 784 and CochStream-large (1.3B parameters) with 24 layers, 16 attention heads, and an embedding size of 2,048. Both models have a vocabulary size of 16,384. The CochStream model takes as input the cochlear token sequence produced by WavCoch and predicts the next token in the sequence. The context length is approximately 20s (4,096 tokens). We utilize a learned positional embedding and compute the cross-entropy loss between the predicted logits and the true next token in the sequence.
>
> We are very happy to supply the actual implementation to the reviewers if they wish, and will publicly release the full code base upon acceptance.
>
> Finally, **we clarify WavCoch’s relation to a mel spectrogram**: WavCoch could operate with a “simple” mel spectrogram as the target instead of a cochleagram. We do not make any claims about the superiority of a cochleagram over a mel spectrogram (we have clarified this in the revised manuscript). Instead, the main novelty from WavCoch lies in encoding one representation (in this case, the waveform) into another representation known to be computed in the auditory processing hierarchy (cochleagram) and “probing” an intermediary representation as discrete tokens. We chose the cochleagram for its biological plausibility.
>
> The cochleagram implementation that we use can be found online in the following public repository, which we will link to in our revised manuscript along with a brief description: https://github.com/jenellefeather/chcochleagram.
>
> **Weakness 2. Issue: Linear probing performance lacks relevant models.**
>
> We agree that smaller models would be helpful. We are currently working on including the models the reviewer suggested (WavLM-base or wav2vec2-base) for comparison the in linear probing results. Following these results, we will also provide a more thorough analysis of our model’s performance by the end of the rebuttal period.

---

> ### Author Response · Authors · 2024-11-15
> **Response to all questions requiring clarification or additional information part II.**
>
> **Questions 1 and 2. Issue: Lacking information about CochStream embeddings.**
>
> We apologize for the oversight. Here is a description, which we have of course included in the revised manuscript:
>
> We obtain CochStream embeddings by pooling the embeddings of all the tokens associated with the corresponding temporal section of the cochleagram via ground-truth phoneme or word boundaries (Section 3.1). For the pooling operation, we tested mean/max/min pooling for the linear probing experiments and lexical similarity for CochStream and the comparison models.
>
> Finally, to answer the reviewer’s question related to how **obtaining embeddings from GPT-style models like CochStream relates to masked models like HuBERT**:
>
> When analyzing a sound (such as when we obtain embeddings for a given phoneme/word), all computations are performed in parallel (using a causal mask, like during training) eliminating any inference time bottlenecks caused by the causality constraint.
> We note that our model is of course also capable of generating long-form continuations of audio, which does take longer to rollout (much like an LLM) – however this is a capability that is not present in competitor models such as HuBERT leaving no clear point of comparison.
>
> **References** \
> [1] Cooley, James W., and John W. Tukey. "An algorithm for the machine calculation of complex Fourier series." Mathematics of computation 19.90 (1965): 297-301. \
> [2] Yu, Lijun, et al. "Language Model Beats Diffusion--Tokenizer is Key to Visual Generation." arXiv preprint arXiv:2310.05737 (2023). \
> [3] Feather, Jenelle, et al. "Model metamers reveal divergent invariances between biological and artificial neural networks." Nature Neuroscience 26.11 (2023): 2017-2034.\
> [4] Radford, Alec. "Improving language understanding by generative pre-training." (2018).

---

> > ### Author Response · Authors · 2024-11-28
> >
> > Thanks again for the time spent reviewing our work. We have now addressed all of your questions and concerns through the individual response (posted last week) as well as through the global response. If you have any further questions that are preventing you from raising our score, please let us know and we will do our best to address them in the extended discussion period.

---

> ### Author Response · Authors · 2024-12-01
> **Approaching Deadline**
>
> We would once again like to thank the reviewer for their comments and suggestions. We have addressed each one of the concerns raised by the reviewer either through a text response or with follow-up experiments.
>
> Specifically, we:
> - Added a diagram depicting architectural details of our pipeline, as well as a detailed description (See Appendix I.1).
> - Added additional base models to compare against CcohStream-base (see Tables 1 and 2). We further discussed advantages and disadvantages of our model (see Section 4).
> - Added information to the “2.1.4 OBTAINING COCHSTREAM EMBEDDINGS” section from our manuscript as it was erroneously left blank in the initial submission.
>
> Since it is getting close to the end of the rebuttal period and we believe that we have addressed all of the stated weaknesses we kindly ask the reviewer to let us know if there are any further unaddressed concerns–otherwise we would be grateful if you could adjust the score accordingly.

---

### Author Response · Authors · 2024-11-28
**Global Response**

We once again thank the reviewers for their constructive and thoughtful feedback which has significantly improved the paper over the rebuttal period. We begin by outlining the key strengths of the paper as identified by the reviewers, followed by a list of the identified suggestions/concerns and the steps we have taken to address them.


**Strengths:**

- Interesting and novel biologically-inspired speech representation learning framework (mentioned by **ipew**, **npk7**, **6c7S**, **Nhso**. For example, **Nhso** noted: “The paper takes an interesting approach by introducing a cochlear-inspired representation (cochleagram) for speech modeling, which is relatively uncommon in the field”)

- Improved interpretability of speech representations (mentioned by **ipew**, **npk7**, **Nhso**. For example, **npk7** noted: “The model can visualize predictions in cochleagram form, offering interpretable insights into its speech representations”)

- Competitive and versatile task performance (mentioned by **ipew**, **npk7**, **6c7S**, **Nhso**. For example, ipew noted: “The authors validate the model’s versatility across various tasks (phoneme/word decoding, lexical semantics) and benchmarks (e.g., SUPERB), demonstrating its competitive performance and interpretability advantages over existing models”)

- Clear writing and good high-level motivation (mentioned by **6c7S**, **Nhso**. For example, **6c7S** noted: “The paper is well-written, with a clear and interesting motivation rooted in mimicking the human auditory system”)

---

> ### Author Response · Authors · 2024-11-28
>
> **Improvements To Address Concerns:**
>
>
>
> **Open-source data reproduction**
>
> We thank reviewer **Nhso** for their suggestion, and in line with our strong commitment to open science, we have reproduced key results using a model trained on fully openly available datasets. First, we retrained a version of WavCoch using the librispeech960 dataset consisting of 960 hours of read English speech [1]. Second, we train a 1B parameter version of the sequence-to-sequence CochStream model on the full 60k hours of the libri-light dataset consisting of 60,000 hours of unlabeled English speech [2]. As demonstrated here, we show that this model demonstrates strong performance on a similar set of tasks than the version trained on our dataset (see Section below titled “CochStream Libri-Light task performance analysis”). This model serves as a strong replication of the results from our initial model using popular open-source datasets.
>
>
> **WavCoch vocabulary size analysis**
>
> As suggested by reviewers **Nhso** and **npk7**, we performed ablations on the vocabulary size of the WavCoch model. In an attempt to ensure maximal reproducibility, we perform these ablations using the librispeech960 dataset as mentioned above. We train variants of WavCoch using a vocabulary size of 16384, 8192 and 4096 (14, 13 and 12 bit codes respectively). We report the results in Figure 3 (https://anonymous.4open.science/w/cochstream-project-page-0546/) which shows the phoneme cluster purity and reconstruction error on an out of distribution test set (TIMIT test set). We introduced the cluster purity metric in our original response (https://openreview.net/forum?id=TQdg1X6eqm&noteId=7OxgsydgYp) which is defined as purity = (Count of most associated phoneme for token i) / (total counts for token i) which intuitively provides a metric for how consistently a given token aligns with a specific phoneme.
>
> The plot shows that a vocabulary size of 8192 presents both the highest cluster purity and lowest reconstruction error. Furthermore, this vocabulary size presents a local minima in the MSE space, and a maxima in the cluster purity space indicating it is the optimal size for generalization in this domain. We are very grateful to the reviewers’ suggestion as it revealed an improvement in our model design. We will use a vocabulary of 8192 for all future models.
>
> **Vocabulary token clustering analysis**
>
> As suggested by reviewers **Nhso** and **npk7**, we analyzed the distribution of tokens associated with specific phonemes in the TIMIT test set. We visualized the distribution in Figure 4 ( https://anonymous.4open.science/w/cochstream-project-page-0546/)  and found that multiple tokens are typically associated with each phoneme. Notably, a large number of tokens are linked to the "sil" (silence) label, but we would like to clarify that this label includes any non-speech sound, not true silence. Since WavCoch is optimized with a single objective–to predict the cochleagram as best as possible–it has no direct incentive to collapse the codebook by merging codes associated with the same phoneme. However, our analyses show that this clustering takes place naturally, to an extent. In future work, we plan to investigate WavCoch models that explicitly incorporate a more efficient codebook, and compare it against the current baseline.

---

> ### Author Response · Authors · 2024-11-28
>
> **Mel spectrogram vs. cochleagram**
>
> As raised by reviewers **ipew** and **Nhso**, we trained a version of our WavCoch model using a standard 80 mel-spectrogram representation (instead of a cochleagram representation). We trained it on the publicly available librispeech960 dataset, consisting of 960 hours of speech recordings [1]. Since the spectrogram L2 reconstruction error is not directly comparable between a cochleagram and mel-spectrogram we utilize two proxy measures: i) Number of unique codes utilized and ii) Phoneme cluster purity. Both of these metrics are computed on the out-of-distribution TIMIT test set.
> First, related to the number of unique codes utilized: We find that the WavCoch model trained with mel spectrograms utilized 8151 out of 8192 codes, while the cochleagram version utilized 8172 codes. Second, related to phoneme cluster purity: We find that the mel spectrogram model achieved an average phoneme cluster purity of 0.3473 while the model trained with the cochleagram achieved an average phoneme cluster purity of 0.3517. While these results are preliminary, they suggest that the cochleagram representation performs at least as well as, if not slightly better than, the mel spectrogram in this context. We are continuously exploring this comparison further, and will report the findings in the final version of the manuscript.
> Finally, besides the quantitative analyses reported above, we prefer the cochleagram over the mel-spectrogram representation for conceptual reasons: The ultimate goal of our framework is to move towards more biologically plausible speech models, and the cochleagram is more aligned with this goal.
>
> In the revised paper, we have added a discussion on the mel-spectrogram versus cochleagram issue and explicitly note that we do not claim the superiority of cochleagrams over mel-spectrograms.
>
>
>
> **CochStream Libri-Light task performance analysis**
>
> As raised by reviewer **Nhso**, we trained a version of CochStream using the 60k hour Libri-Light dataset [2] (denoted as CochStream-large-ll in the updated version of the paper). We utilized cochlear tokens produced by the new WavCoch model with a vocabulary size of 8192 which proved superior in the experiments mentioned above. We also adjusted the CochStream Transformer parameters to exactly match those of HuBERT (and WavLM) for an even more apples-to-apples comparison (suggested by e.g., reviewer ipew). While our full training schedule is set for 500k optimization steps, we were not able to complete it in time for rebuttal due to time and computational constraints. Nevertheless, even at 200k steps our model already matches the performance of the model we described in the initial submission. We highlight some of the results below:
>
> On the TIMIT linear probing task the libri-light model matches the performance of the one trained on our data on phoneme decoding (92% old vs. 92% new) while it outperforms it on the word decoding task (67% old vs. 69% new).
> On the ZeroSpeech 2021 Lexical Semantic Benchmark the libri-light model scored above the old base model and all other baselines we evaluated, but below the large model trained on our dataset. This suggests that for this task, both our modeling approach and our dataset are important for maximizing the performance.
> On the SUPERB benchmark the libri-light model faired comparably to the model trained on our data.
>
> For the full results please see tables 1, 2 and 3 in the updated main paper pd.
>
> We believe that these results demonstrate that the core performance contribution of our work comes from our novel two-stage modeling framework (WavCoch tokenization followed by CochStream auto-regressive prediction). Furthermore, this demonstrates that our model is not contingent on one single dataset – we obtain very competitive performance on different audio datasets. We are continuously evaluating our models on SUPERB, and will send a further update in the extended discussion period about performance characteristics as suggested by reviewers.

---

> ### Author Response · Authors · 2024-11-28
>
> **Interpretability of speech continuation capability**
>
> As suggested by reviewers **npk7** and **6c7S**, we investigate the decoding of CochStream predictions back into the auditory signal. To this end, we developed a simple procedure for inverting the cochleagrams back into a waveform. This procedure is a per-sample optimization of making the waveform match the cochleagram prediction.
>
> Specifically, we optimize a 1D tensor representing the waveform input to make its cochleagram representation match the cochleagram predicted by WavCoch (via L2 error). We backpropagate through the cochleagram transformation and use the Adam optimizer with a learning rate of 1e-2.
> Note that this optimization procedure is not a learned vocoder model, but a simple procedure of converting the output of WavCoch, the cochleagrams, into audible sound (conceptually similar to Griffin-Lim algorithm [3]).
>
> We upload several audible samples of speech generations from our CochStream: (https://anonymous.4open.science/w/cochstream-project-page-0546/). Please access the page using Google Chrome or Firefox as we have seen some cases of Safari not properly loading these videos. We truly hope that the reviewers will take a few seconds to listen to the continuations.
>
> We observe that on short time-scales the model produces reasonable completions, but the longer the completion, the more the predictions drift away from being plausible. We would like to emphasize that the purpose of CochStream is not to be a language model, but a speech representation model – the fact that it can perform rudimentary language modeling is a serendipitous side effect of the training objective, which points to the fact that understanding speech, and producing language can be thought of as a unified objective. These findings serve as great motivating factors for a planned follow-up work which will attempt to stabilize longer term speech generations, building on top of the foundation laid out in this paper.

---

> ### Author Response · Authors · 2024-11-28
>
> **References:**
>
> [1] Panayotov, V., Chen, G., Povey, D., & Khudanpur, S. (2015, April). Librispeech: an asr corpus based on public domain audio books. In 2015 IEEE international conference on acoustics, speech and signal processing (ICASSP) (pp. 5206-5210). IEEE.
>
> [2] Kahn, J., Riviere, M., Zheng, W., Kharitonov, E., Xu, Q., Mazaré, P. E., ... & Dupoux, E. (2020, May). Libri-light: A benchmark for asr with limited or no supervision. In ICASSP 2020-2020 IEEE International Conference on Acoustics, Speech and Signal Processing (ICASSP) (pp. 7669-7673). IEEE.
>
> [3] Griffin D. and Lim J. (1984). "Signal Estimation from Modified Short-Time Fourier Transform". IEEE Transactions on Acoustics, Speech and Signal Processing. 32 (2): 236–243. doi:10.1109/TASSP.1984.1164317

---

### Meta-Review · Area_Chair_KT8B · 2024-12-26

**Metareview:**

The paper introduces CochStream, a "biologically inspired framework" for speech representation learning based on cochlear-inspired processing. It comprises two stages -- an encoder converts audio into cochleagrams -- a biologically motivated representation -- including a LFQ layer to generate discrete tokens, and a GPT-style autoregressive model for token prediction. The paper evaluates the model on SUPERB tasks (e.g., phoneme recognition, word decoding, lexical semantics), showing competitive or superior performance to existing baselines.

**Strengths** (1) The cochlear-inspired approach introduces an interesting, interpretable representation that aligns with human auditory processing, contributing a unique perspective to the field. (2) Competitive performance, especially in lexical semantic similarity, is demonstrated on standard benchmarks.

**Weaknesses** (1) Reviewers pointed out the current form lacks necessary ablation studies on both representations, training strategies, and vocabulary sizes. Additionally, a comparison with SoTA models trained on the same dataset is also requested. (2) Some reviewers raised concerns about the clarity of the model design, which makes it hard to understand. (3) The contribution/advantage of biological claims is not clear.

**Decision** This paper received mixed reviews with two reviewers not engaging in the discussion phase. This creates difficulty in making the final decision. It is clear that the authors made extra efforts to include additional experiments and comparisons based on the reviewers' request, however, there remain concerns between reviewers (e.g. npk7) until the end of the discussion phase. Therefore, I am leaning toward rejection based on the current version.

**Additional Comments On Reviewer Discussion:**

The discussion mainly happened with reviewer Nhso where the author provided additional clarification and details according to each weakness the reviewer had pointed out, as well as the follow-up questions on novelty claims. Also, the manuscript was updated based on the request. The reviewer improved the original rating to borderline acceptance.

While npk7 added one question during the rebuttal phase, npk7 did not acknowledge or change the scores based on the authors' responses. This indicates the new experiments may not fully address the reviewer's concerns.

Both the other two reviewers did not engage with the author about their questions, while the authors included additional experiments into the updated manuscript to reflect the request.

---

### Decision · Program_Chairs · 2025-01-22

Reject